# The Impact of MOSE (Experimental Electromechanical Module) Flood Barriers on Microphytobenthic Community of the Venice Lagoon

**DOI:** 10.3390/microorganisms11040936

**Published:** 2023-04-03

**Authors:** Laura Baldassarre, Vanessa Natali, Fabio De Pascale, Alessandro Vezzi, Elisa Banchi, Matteo Bazzaro, Federica Relitti, Davide Tagliapietra, Tamara Cibic

**Affiliations:** 1Oceanography Section, National Institute of Oceanography and Applied Geophysics—OGS, 34010 Trieste, Italy; 2Department of Biology, University of Padua, 35122 Padua, Italy; 3Institute of Marine Sciences, National Research Council, CNR-ISMAR, 30122 Venezia, Italy

**Keywords:** microphytobenthos, benthic diatoms, Venice Lagoon, next generation sequencing, classic taxonomy

## Abstract

MOSE is a system of mobile gates engineered to temporarily isolate the Venice Lagoon from the Adriatic Sea and to protect the city from flooding during extreme high tides. Within the framework of the Venezia2021 program, we conducted two *enclosure* experiments in July 2019 (over 48 h) and October 2020 (over 28 h) by means of 18 mesocosms, in order to simulate the structural alterations that microphytobenthos (MPB) assemblages might encounter when the MOSE system is operational. The reduced hydrodynamics inside the mesocosms favored the deposition of organic matter and the sinking of cells from the water column towards the sediment. Consequently, MPB abundances increased over the course of both experiments and significant changes in the taxonomic composition of the community were recorded. Species richness increased in summer while it slightly decreased in autumn, this latter due to the increase in relative abundances of taxa favored by high organic loads and fine grain size. By coupling classical taxonomy with 18S rRNA gene metabarcoding we were able to obtain a comprehensive view of the whole community potential, highlighting the complementarity of these two approaches in ecological studies. Changes in the structure of MPB could affect sediment biostabilization, water turbidity and lagoon primary production.

## 1. Introduction

With the climate change effects intensifying, extreme events are becoming more and more frequent, severe, and unpredictable, and sea level is predicted to rise between +0.6 and +1.2 m within the end of this century [1]. Coastal and lagoonal areas are among the most vulnerable environments to these derangements, but also among the most valuable ones on Earth, providing a range of ecological, cultural and socioeconomic services [2,3]. The Venice Lagoon (northern Adriatic) represents an important study case for both socio-economic interest and ecosystem preservation, which has undergone considerable changes in the last 30 years [4]. In the last century, the lagoon has been subjected to several anthropogenic modifications, and, since 2004, the morphology of the lagoon inlets has been strongly modified by the construction of MOSE (Italian acronym for “Experimental Electromechanical Module”) (https://www.mosevenezia.eu, accessed on 23 March 2023), with consequent alterations on the exchange of fine sediments between the inner basin and the sea [5,6]. MOSE is a system of mobile barriers (Figure 1F) positioned at each of the three inlets of the Venice Lagoon, built to prevent flooding of Venice City. Under normal conditions, the barriers lie on the seabed, and are raised when flood events (tide above 110 cm) are predicted. Since the MOSE start-up, the average closures have lasted from a few hours to 24 h and the very high tide conditions have occurred four times per year (Venice City Hall website www.comune.venezia.it, accessed on 23 March 2023). According to [7,8], it is predicted that, in the near future, the barriers will have to be closed more than 200 times and for 1000 h per year. Since its construction, the MOSE system has led to important modifications to the lagoon [6,9], and its increasingly frequent operating times will progressively exacerbate the lagoon’s transformation, implying further changes in the hydrodynamics [10], the lengthening of the water renewal period throughout the lagoon [11], and the alteration of sediment fluxes, pollutants and organisms [12,13] and wetland morphodynamics [14]. Moreover, if the closure occurs during the warm season, with high water temperatures and low hydrodynamic conditions, the higher organic load may lead to more persistent and severe hypoxic/anoxic events [15], with severe consequences for the lagoonal ecosystem. In a recently published study [16], the authors reported that, in sites confined by the MOSE closure, sediments exert an important depleting action on dissolved oxygen already at water temperatures just above 20 °C, which are not particularly high for the lagoon.

The microphytobenthos (MPB) constitute an important component of the benthic ecosystem. Indeed, this community contributes to the total production, plays a crucial role in the benthic food web and in sediment stabilization through the production of extracellular polymeric substances (EPS), and regulates nutrient fluxes and other biogeochemical processes at the sediment interface [17,18,19,20,21]. In shallow coastal systems, benthic microalgae, and diatom assemblages in particular, contain considerable ecological information and are therefore used as indicators of various types of stress: nutrient loads, organic enrichment, and hypoxic conditions [22,23,24,25]. As the main oxygen producer in the absence of macroscopic vegetation, MPB enables aerobic degradation of organic matter in sediments; yet, in terms of community structure, their response to hypoxic events in coastal marine lagoons is still rarely studied [24,25,26,27]. The first investigation on the distribution and diversity of MPB in surface sediments of the Venice Lagoon was carried out in 1994–1995 by [28]. Since then, a large number of taxonomic, ecological and modeling studies have been conducted [18,29,30,31]. In this study, for the first time, we applied in situ mesocosm experiments in the Venice Lagoon to investigate the impact of the MOSE infrastructure on the MPB community.

In the last two decades, DNA metabarcoding has been more and more widely used in addition to traditional taxonomic studies, especially on microbial communities. Nowadays, the available sequencing techniques and databases are extremely advanced and constantly upgraded. Nevertheless, it is clear to taxonomists that classical taxonomy still cannot be fully replaced by molecular tools; instead, the two approaches are complementary and provide different but equally important information [32,33,34].

Assessing the impacts that new infrastructures might have on the ecosystems of interest is of pivotal importance in order to establish effective and sustainable protection strategies. Within the framework of the “Venezia2021’’ project, the present study was part of the enclosure experiments aimed at monitoring the MOSE impact on the Venice Lagoon ecosystem. A great effort was made in order to set up and run such challenging in situ experiments in an unstable and sensitive environment such as the Venice Lagoon. By simulating the MOSE closure, these experiments aimed at identifying structural and functional alterations in the planktonic and benthic populations. In this work, we focused on the effects of the altered hydrodynamics induced by the operational phase of the MOSE system on the MPB community. In particular, we investigated the modifications in MPB abundance and composition, the latter through both operator-driven classical microscopy (classical taxonomy) and NGS sequencing. This way, we were able to obtain a more comprehensive and detailed overview than would have been possible by using either approach alone, and to highlight pros and cons of both techniques.

We hypothesized that the reduced hydrodynamics and oxygenation of the water column and sediment portions enclosed by the barriers induce modifications in the MPB total abundance and structure. Our guiding questions were: (1) Does the particular matter settling from the water column towards the sediment have a stimulatory effect, leading to an increase in MPB absolute abundances? (2) Do these modified hydrodynamic conditions affect the MPB structure, resulting in an increase of opportunistic species? Furthermore, to investigate the MPB community structure, we compared two methodologies, i.e., classical taxonomy and the more innovative 18S rRNA gene metabarcoding. Therefore, our third question was: (3) Are the higher resolution and efficiency of the molecular tools preferable over the time-consuming but strictly operator-driven classical microscopy?

In this comparison, Cyanobacteria were not included since we applied primers to target only microalgae, but they were kept in the presentation of the taxonomic results because prokaryotic photosynthetic organisms are considered part of the MPB community [35].

We expected to obtain a wider taxonomic characterization from the 18S rRNA gene sequencing, but to obtain a more reliable quantification of the living and photosynthetically active taxa of the MPB community from the classical taxonomy. Finally, we briefly discussed the ecological implications of the modified MPB community structure on sediment biostabilization and lagoon ecosystem functioning.

## 2. Material and Methods

### 2.1. Study Area

The Venice Lagoon is located at the north of the Adriatic Sea (Figure 1A) and is the largest Mediterranean lagoon, with a total surface of 550 km^2^, a mean width of 15 km and an average depth of 1.5 m [12]. It is characterized by a network of channels, flats and shoals, and the bottom sediments consist mostly of clay and silt inside the lagoon, but are sandy in the proximity of inlets [36]. The Venice Lagoon is exposed to two major wind events: the scirocco, an autumnal/spring wind that blows from the southeast, and the bora, which prevails in winter and blows from the northeast. Three inlets (Lido, Malamocco and Chioggia, from north to south) allow water exchanges with the Adriatic Sea, which are driven by a semidiurnal tide with an excursion of about 1 m in spring conditions, and the residence time varies between 24 h close to the inlets and 30 days in the internal lagoon [9].

In this study, a shallow (<75 cm) non-anthropized area, Palude di Cona, with organic enrichment and low hydrodynamics, and subjected to hypoxic events, in the northernmost part of the basin, close to the bayhead estuaries of some plain rivers (the Sile, Dese and Zero), was chosen for the experiments. This area is a typical brackish area, surrounded by salt marshes with an average depth of 0.8 m during the mean tidal conditions [37], and preserving different estuarine habitats including mud flats, salt marshes and tidal channels [4,12]. The conspicuous amount of sediment in this area is the result of rapid settling of suspended particles of freshwater origin deriving from the Silone branch of the Sile River, and the Dese and Zero Rivers, accounting for about 46% of the total riverine inputs of suspended matter to the lagoon [38] (Figure 1A).

### 2.2. Experimental Design

The mesocosm experiments were carried out in Palude di Cona, near the Dese River mouth (Figure 1A). The first experiment was performed between 24 and 26 July 2019, and the second one between 13 and 14 October 2020.

A total of 18 mesocosms (~0.8 m^3^ each) consisting of a cylindrical zinc-galvanized iron frame (∅ ca. 100 cm) coated by a transparent nylon cylinder (∅ ca. 106 cm) (Figure 1D,E), were aligned in three rows (corresponding to three experimental replicates: R1 = 45.510365° N, 12.401300° E; R2 = 45.510259° N, 12.402228° E; R3 = 45.511406° N, 12.401957° E) along transects with similar depth (<75 cm) (Figure 1B,C). Since during positioning them a navigation canal with higher bathymetry had to be avoided, they could not be equally distanced; the distances between the centroids of the clusters were the following: R1–R2 = 75 m; R1–R3 = 125 m and R2–R3 = 130 m, whereas the individual mesocosms were 5 m apart one from another. The mesocosms were open at both the lower and upper sides (Figure 1E). In this way, vertical fluxes between the two boundary layers (water/air and water/sediment) were allowed, whereas the horizontal ones (water between the inside and outside of the mesocosms) were blocked, mimicking the segregation of the lagoon from the oxygen-enriched seawater when the flood barriers are raised for several hours (Figure 1F).

At each sampling, water temperature, salinity and dissolved oxygen concentration were recorded by a multiparameter probe EXO2, YSI, Xylem Inc., USA, prior to sediment sampling, inside and outside the mesocosms.

Virtually undisturbed sediment samples were taken by a hand-operated sediment corer using polycarbonate sample tubes (HAPS Model 50.820, KC Denmark A/S, Denmark, sample area = 127 cm^2^, sample depth = 31 cm). At the first sampling time (T0), sediments were sampled outside the three replicate mesocosms, in close proximity to them. Then, the cylindrical nylon bags of the 18 mesocosms, initially lowered to allow water exchange, were pulled up, and the experiment began. At each following sampling time (T1–T4), sediments were collected from inside the three replicate mesocosms, one from each row, according to a chronology aimed at identifying any responses of the induced closure on the benthic microalgal community in the short term (after 4 h from the positioning of the structures = T1) and in the long term (24 h = T2, 28 h = T3 and 48 h = T4), starting from an initial state (T0). Once in the laboratory, the sediment cores were extruded and the top sediment layer (0–1 cm) was collected for chemical (Total Organic Carbon—TOC, Total Nitrogen—TN, grain-size, Biopolymeric Carbon—BPC) and microbiological analyses (microphytobenthic abundance and community composition). The field experiment lasted 48 h in July 2019, and 28 h in October 2020 due to the high tide and the consequent actual activation of the MOSE in the second experimental day, which forced us to stop the second experiment in advance. In the first experiment, one of the three replicates at T4 was lost due to a logistic issue. Notwithstanding, each mesocosm enclosed a sediment area of 0.785 m^2^, quite a representative portion of the lagoon; therefore, two replicates were still depictive enough.

### 2.3. Physical and Chemical Analyses

Grain size analysis was carried out on bulk samples (10–25 g) collected from each mesocosm and sampling time, sieved at 2 mm and pretreated with 10% hydrogen peroxide before being analyzed with an LS 13 320 Laser Diffraction Particle Size Analyzer, Beckman Coulter, USA. Data are expressed as sand, silt, and clay percentages following the Udden-Wentworth grain-size classification [39].

For TOC, TN and BPC analyses, sediment was freeze-dried, ground in a ceramic mortar and then sieved through a 250 μm iron steel sieve (Endecotts Ltd., UK).

For TOC and TN determination, triplicate subsamples of about 8–12 mg were weighed directly into silver and tin capsules using a microultrabalance with an accuracy of 0.1 μg. Before TOC analysis, subsamples were treated directly into capsules with increasing concentrations of HCl (0.1 N and 1 N) to remove the carbonate fraction [40]. C and N contents were determined using a CHNS-O elemental analyzer ECS 4010, NC Technologies, Italy, according to [41]. Standard acetanilide (Costech, purity ≥ 99.5%) was used to calibrate the instrument, and empty capsules were also analyzed in order to correct for blank. Quality control of measurements was performed using internal standards and it was also verified for carbon against the certified marine sediment reference material PACS-2 (National Research Council Canada). The relative standard deviations for three replicates were lower than 3%. TOC and TN concentrations were expressed as weight percentage of C and N, respectively, on dry sediment.

Subsamples were processed in triplicates for the determination of carbohydrates, lipids and proteins. Colloidal and EDTA extractable carbohydrates (C-CHO-H_2_O and C-CHO-EDTA) were analyzed following the method described by [42]. Lipids (C-LIP) were analyzed following the method proposed by [43] and modified for sediments. Proteins (C-PRT) were extracted in NaOH (0.5 M) for 4 h and determined according to [44]. The concentrations of C-CHO, C-PRT and C-LIP were expressed as glucose, bovine serum albumin and tripalmitine equivalents, respectively. Data were converted to carbon equivalents using the conversion factors proposed by [45,46]: 0.49 g C g^−1^ for C-CHO, 0.50 g C g^−1^ for C-PRT and 0.75 g C g^−1^ for C-LIP, and the sum of carbohydrate, lipid and protein carbon was referred to as BPC sensu [45].

### 2.4. Abundance and Community Structure of Microphytobenthos Using Classical Taxonomy

In this study, by the term microphytobenthos (MPB), we refer to the microscopic eukaryotic algae (diatoms, dinoflagellates, flagellates, etc.), and prokaryotic photosynthetic organisms, such as filamentous cyanobacteria. For MPB analyses, from each mesocosm replicate, three aliquots of homogenized sediment (2 cm^3^) were withdrawn using a syringe and directly fixed with 10 mL of formaldehyde (4% final concentration)-buffered solution CaMg(CO_3_)_2_, in pre-filtered bottom seawater (0.2 μm filters). After manual stirring, 20 μL aliquots of the sediment suspension were drawn off from the slurries and placed into a counting chamber. Only cells containing pigments and not empty frustules were counted under a Leitz inverted light microscope (Leica Microsystems AG, Wetzlar, Germany) using a 32X or 40X objective (320X or 400X final magnification) [47]. For each mesocosm, two (when the standard deviation did not exceed 15%) [48] or three replicates were counted. When possible, at least 200 cells were counted per sample to evaluate also rare species. The microalgal taxonomy was based on the AlgaeBase [49] and WoRMS [50] websites. The qualitative identification of MPB assemblages was carried out using the [51,52,53,54,55,56,57,58,59,60] identification keys, as well as identification keys of freshwater microalgae [61].

### 2.5. Benthic Microeukaryotic Community Composition through Metabarcoding

For 18S rRNA gene metabarcoding analysis, sediment from each mesocosm replicate of T0 and T final (T4 for Summer 2019 and T3 for Autumn 2020) was sampled with a sterile spatula and stored at −80 °C until further processing.

DNA was extracted from each sample using the DNeasy PowerSoil Pro Kit (Qiagen) following the manufacturer’s instructions and quantified with a Qubit Fluorimeter (Thermo Fisher Scientific, Waltham, MA, USA). The samples were then amplified with degenerated primers covering the V9 region of the 18S rRNA gene from position 1389 to 1510 (forward—TTGTACACACCGCCC; reverse—CCTTCYGCAGGTTCACCTAC) [62]. PCR mixtures, in a final volume of 25 μL, were the following: 5 ng of template DNA, 0.4 U of Phusion™ High-Fidelity DNA Polymerase (Thermo Fisher Scientific, USA), 1X Phusion HF buffer, 0.5 μM of each primer and 200 µM of each dNTP. PCR amplifications (98 °C for 4 min; 25 cycles of 98 °C for 20 s, 57 °C for 30 s, 72 °C for 30 s; 72 °C for 5 min) were set up in triplicate in order to smooth possible intra-sample variance. PCR products were visualized on 1.5% agarose gels, then amplicon triplicates were pooled and purified using 0.65X volumes of AMPure XP beads (Beckman Coulter, Brea, CA, USA).

The pooled PCR products were then indexed and subsequently normalized according to the “16S Metagenomic Sequencing Library Preparation’’ protocol (Illumina Inc., Hayward, CA, USA), with one main modification: the PCR amplicons were normalized using the SequalPrep Normalization Plate kit (Thermo Fisher Scientific, USA). Finally, amplicon libraries were equally pooled, purified using 0.7X volumes of AMPure XP beads (Beckman Coulter, USA) and sequenced in two separate runs (for 2019 and 2020 samples) of a MiSeq Illumina (2 × 300 bp). Primers were removed using cutadapt, v.2.1 [63]. The analyses were carried out using the R software environment for statistical computing [64] and the phyloseq package [65]. Low-quality filtering was performed by trimming the sequences to 120 bp. The remaining bases still allowed good amplicon overlap. The sequences were then processed with dada2, v.1.20, to denoise and remove errors [66]. The taxonomic assignment of the sequences was performed using the Protist Ribosomal Reference (PR2) v.4.14 databases [67] and unassigned sequences were removed. After these purging steps, rare and low-abundant ASVs were filtered: features with abundances across all sediment samples lower than 10 total counts and present only in less than four samples were removed. The remaining ASVs were kept for the following analysis. The raw data are deposited at the Sequence Read Archive (SRA) and available under the project PRJNA915329.

### 2.6. Statistical Analyses

An independent t-test was performed between the water temperature, salinity and dissolved oxygen concentration recorded inside the mesocosms at the beginning and at the end of the experiments. A preliminary Spearman’s correlation analysis was performed through Past 4 [68] in order to select the abiotic parameters more related to the biotic patterns. All further statistical analyses were performed using PRIMER 7.0.21 [69]. For each experiment, two biotic matrices were constructed, based on the MPB abundances (Appendix A): one for univariate analysis (k-dominance and diversity indices) considering all taxa at the genus and species levels, and one for multivariate analysis (nMDS and BIO-ENV), in which also higher taxonomic levels were taken into account. In October 2020, the matrix was also cleaned by eliminating the epiphytic species *Navicula rhombica*, i.e., a tube-dwelling diatom whose abundance in samples might be random or often uncountable. An additional matrix (Appendix A) was constructed and normalized for diversity analysis with the metabarcoding data at the ASV level. Diversity analysis was applied to MPB data, i.e., abundances at the genus and species levels obtained by classical microscopy or ASV level obtained through molecular tools, considering richness (d [70]), equitability (J′ [71]), and diversity (H’(log_e_) [72]) and dominance (λ [73]).

Before multivariate analyses, the biotic matrices were square-root transformed and Bray–Curtis similarity matrices were generated. To visualize differences in taxa assemblages among the different sampling points, a non-metric multidimensional scaling ordination (nMDS) [74] was performed on MPB. To highlight which taxa mainly contributed to the temporal variation of the assemblages, the taxa with the highest (average ≥ 5%) RA were overlaid on the nMDS plot. The normalized (z-standardization) environmental variables (TOC, sand, clay, BPC: water soluble carbohydrates, EDTA extractable carbohydrates, proteins and lipids) were fitted as supplementary variables (vectors) onto ordination spaces to investigate their effects on community structure; TN and silt were omitted from the analyses because they were correlated with TOC and sand, respectively. Clusters among groups were inferred and plotted by applying the SIMPROF (Similarity Profile) analysis. In order to test which environmental variables correlated best with the patterns of taxa according to the experimental time point, a Euclidean distance matrix was constructed on the physical-chemical data (TN, TOC, BPC and grain-size fractions) and a BEST (BIO-ENV + STEPWISE) analysis based on Spearman’s coefficient [75] was performed.

Comparison among experimental times was complemented by a visual representation of diversity, using k-dominance curves [76]: species abundances (average of three mesocosms) were ranked (in log) in decreasing order of dominance and plotted cumulatively. Taxa and ASVs differentially represented at the beginning (T0) or at the end (T4 or T3) of the two experiments were identified by LDA Effect Size (LEfSe) [77]. LEfSe uses the non-parametric factorial Kruskal–Wallis sum-rank test to detect features (species in this case) with significant differential abundance, concerning the biological conditions of interest (i.e., experimental time point); subsequently, it uses Linear Discriminant Analysis (LDA) to estimate the effect size of each differentially abundant feature.

## 3. Results

### 3.1. Water Column Parameters

Prior to sediment sampling, water temperature, salinity and dissolved oxygen were recorded by a multiparameter probe, inside and outside the mesocosms (at the beginning of the experiment = T0, outside and inside coincided) (Appendix A). No significant variation was recorded for temperature and salinity, while the dissolved oxygen concentration displayed a slow significant decrease over the first experiment inside the mesocosms compared to the outside (*t*-test: R = −2.81, *p* < 0.05).

### 3.2. Microphytobenthic Community Using Classical Taxonomy

In July 2019, the total MPB abundance ranged from 39,000 ± 1742 cells cm^−3^, observed in one replicate of T0, and 137,600 ± 4064 cells cm^−3^ obtained in one replicate of T1 (Figure 2A). The average abundance of the replicates was the lowest at T0 (48,200 ± 12,019 cells cm^−3^), while the highest was recorded at T4 (134,250 ± 4030 cells cm^−3^).

The MPB community was dominated by diatoms, whose abundance varied between 82.07% at T2-R3 and 98.03% at T4-R1.

In October 2020, total MPB abundance varied between 48,200 ± 2970 cells cm^−3^ observed at T2-R1, and 252,600 ± 30,547 cells cm^−3^ at T3-R1. The lowest average abundance (112,400 ± 36,628 cells cm^−3^) of the whole experiment was observed at T1, with the highest (219,667 ± 38,119 cells cm^−3^) at T3 (Figure 2B).

As of July 2019, the MPB community was dominated by diatoms, which represented a variable fraction between 87.14%, obtained at T2-R1, and 98.86% obtained at T2-R2. When diatoms were the least abundant (T2-R1), all other groups, except phytoflagellates, showed their Relative Abundance (RA) maxima.

Considering July 2019, within the Bacillariophyceae class, we identified 26 genera, among which the most abundant were *Tryblionella* and *Thalassiosira*, with RAs of 39.6% and 17.6%, respectively. We identified three genera belonging to Cyanobacteria: *Anabaena*, *Oscillatoria* and *Spirulina*, with the first two being the most abundant at T0 (average RA = 0.7% and 0.6% respectively). We were able to identify only one genus of Chlorophyta, namely *Scenedesmus*, recorded at T0.

In October 2020, we identified 25 genera within the Bacillariophyceae class, among which the most abundant were *Mastogloia*, *Gyrosigma* and *Surirella*, with RAs of 20.26%, 16.40% and 11.72%, respectively. Three genera were identified within the Cyanobacteria: *Anabaena*, *Oscillatoria* and *Merismopedia*, although with relatively low mean abundances of 0.5%, 0.4% and 0.3%, respectively. In the Chlorophyta group, as in July 2019, only the genus *Scenedesmus* was identified, present solely at T2-R1 (average RA = 1.12%).

Focusing on the specific composition of the diatom community in July 2019, the most abundant species on average (RA ≥ 0.5%) was *Tryblionella* cf. *acuminata* (Appendix A). The community-specific composition at T0 (outside the enclosures) was different compared with all the other experimental times. At T0, the most represented species was *Paralia ulcate*; at T0-R2, we recorded the lowest cell number (N) and equitability of the MPB community (J’ = 0.72; Appendix A). On the other hand, the highest value of equitability was found at T2-R2 (J’ = 0.87). At T1-R1, the MPB community was characterized by a greater number of taxa; it showed, in fact, the highest richness (d = 2.90) and diversity (H’ = 2.88) of the whole experiment. On the contrary, T1-R2 displayed the lowest diversity (H’ = 2.16) and the highest dominance (λ = 0.21), due to the abundant occurrence of the genera *Thalassiosira* and *Tryblionella*, reaching RAs of 35.1% and 31.4%, respectively. Among all the experimental times from T1 to T4, we observed a general dominance of *Tryblionella* cf. *acuminata* and *T.* cf. *compressa*; however, moving towards T4, Psammodyction cf. constrictum prevailed, reaching an average RA of 10.7%, slightly higher than that of *Tryblionella* cf. *acuminata*.

In October 2020, the most abundant diatom species (RA ≥ 0.5%) in all experimental time points was *Mastogloia braunii* (Appendix A). The genus *Gyrosigma* was highly abundant and represented by several species during the experimental times, namely: *Gyrosigma macrum*, *G. fasciola*, *G. acuminatum*, *G. spencerii* and *G. tenuissimum*; the last one reached an RA > 0.5 only at T2. At T0-R1, the community displayed the highest diversity (H’ = 3.20), considering both the experiments, and consequently the lowest dominance (λ = 0.06, Appendix A). Conversely, at T1-R3, the community showed the lowest diversity (H’ = 2.82) due to the presence of the genus *Mastogloia*, represented by only a few species. At T2-R1, we recorded the lowest number of cells and species but the highest equitability (J’ = 0.92).

The temporal pattern of biodiversity in both the experiments was analyzed in more detail through k-dominance curves. In July 2019, the typical semi-sinusoidal curves (Figure 3A), which reflect low dominance and high species richness, were observed especially at the end of the experiment, when biodiversity increased due to the occurrence of several species belonging to the genus *Tryblionella*. On the contrary, at the beginning of the experiment, two replicates of T0 and one replicate of T1 showed the typical cut-off form of a community dominated by a few species (Appendix A).

Conversely, in October 2020, the typical semi-sinusoidal curves were observed in all the experimental time points (Figure 3B), indicating no dominance of any particular taxa, as confirmed by diversity indices (Appendix A).

In the nMDS ordination, three distinct groups were inferred by SIMPROF in July 2019 (Figure 4A): the first was constituted by all samples of T0 and was placed far from the other time points, the second was made up of all samples of T4 plus one replicate of each intermediate time point, and the third constituted of other samples from T1 to T3. *Paralia sulcata* was associated with the first group, whereas different *Tryblionella* species with *Psammodyction* cf. *constrictum* were associated with the other groups, in agreement with the pattern of the k-dominance curves (Figure 3A).

Further, we applied the LEfSe analysis to identify which MPB taxa were the most discriminating between the beginning (T0) and the end of the experiment (T4). The taxa *Mastogloia* sp., *Amphora* spp. and *Anabaena* sp. significantly decreased in abundance at T4, while six other taxa i.e., *Psammodyction* cf. *constrictum*, *Tryblionella* cf. *hungarica*, *Nitzschia* cf. *commutata*, *Mastogloia* cf. *cuneata*, *Achnanthes* spp. and *Surirella* spp. increased (Figure 5A).

In October 2020, two groups emerged in the nMDS ordination: one comprising all T0 samples, and the other comprising all T3 samples; one T1 replicate and one T2 replicate were also included at the edges of both groups (Figure 4B). Nevertheless, the group clustering was less defined than in July. The group comprising all T0 samples was mainly influenced by chemical parameters plus the following five species: *Surirella* cf. *robusta*, *G. fasciola*, *Nitzschia fasciculata*, *G. macrum* and *Mastogloia braunii*. The LefSe analysis at T3 determined *Surirella* spp., *G. macrum* and *G. fasciola* together with *Entomoneis* sp. and *Amphora* spp. as the most discriminating taxa. In contrast, *Tryblionella* cf. *apiculata*, *Mastogloia* cf. *acutiscula*, *Biddulphia* spp., *Entomoneis* cf. *alata* and *Gyrosigma* cf. *balticum* displayed significantly higher abundances at T0 than at T3 (Figure 5B).

Furthermore, the BIO-ENV analysis (Table 1) indicated that, in July 2019, the MPB assemblage was best correlated (R = 0.500) with four abiotic variables: sand, clay, TOC (total organic carbon) and C-LIP (lipids); while in October 2020, the abiotic variables that best correlated (R = 0.625) with the MPB assemblage were TN, TOC, C-PRT (proteins) and C-LIP.

### 3.3. Microphytobenthic Community through Metabarcoding

For this study, an average of 591,530 ± 99,617 paired-end sequences were produced for each sample, with a total sequencing depth of over 7 million sequences. After the initial steps of filtering, denoising, merging and chimera removal, an average of 297,516 ± 93,303 sequences per sample were retained; these were used for all subsequent analyses. The rarefaction curves for all the samples had reached a plateau, despite the samples having a different sequencing depth (Appendix A). The initial number of total ASVs was 10,288; of these, 4199 were not assigned to any phylum, 781 were assigned only at the supergroup level (divided as follows: Alveolata (115), Amoebozoa (125), Archaeplastida (43), Excavata (5), Hacrobia (4), Opisthokonta (246), Rhizaria (17), Stramenopiles (226)) and 1100 were assigned to Metazoa. The remaining 4208 ASVs were eligible for subsequent analyses and were reduced to 1066 after filtering for low abundant taxa (see material and methods), representing 38 different phyla. From these ASVs, 444 ascribable to MPB were kept for further analyses. The number of sequences/sample varied between 13 and 16% in the two experimental times of the first experiment and between 16 and 26% in the two experimental times of the second experiment.

The 18S sequencing outcomes confirmed the higher diversity of the MPB community in autumn compared to summer (Figure 6B and Appendix A), with the maximum number of ASVs of the whole study recorded at T0-R3 in October 2020 (S = 329). For the other indices, with the exception of λ, the absolute maximum was also recorded in October 2020 at T0. The lowest dominance was observed in October 2020 (λ = 0.05), confirming a more evenly distributed MPB community than in July 2019, noticeable also from the higher relative number of reads (≅ 40%) of taxa representing < 0.5% of the whole community (Figure 6B). The results evidenced an overall loss of richness (d) and diversity (H’) from the beginning through the end of both the experiments (Appendix A). The molecular analysis was able to detect microalgae non detectable through microscopy (Figure 6A) (i.e., members of the Chlorophyta, Cryptophyta, Prymnesiophyta and Chrysophyta classes) (Figure 6B). The dominance of the genus *Tryblionella* detected in July 2019 using classical microscopy (Figure 6A) was not confirmed by the sequencing data, while *Thalassiosira* constituted up to 27% of the total number of reads (Figure 6B). In October 2020, properly dominant taxa could not be detected from the sequencing data, whereas the not-reliably-classified sequences constituted the majority of the community. It is interesting to note that, among the most abundant taxa of the MPB community, we also detected some belonging to the planktonic forms (i.e., *Melosira, Chaetoceros, Minutocellus* and *Thalassiosira*) and phytoflagellates. LEfSe detected 21 and 44 ASVs overrepresented at T0 compared to T4 and T3 in July 2019 and October 2020, respectively. On the other hand, 8 and 12 ASVs, respectively, were overrepresented at T4 and T3 compared with T0 of the two experiments (Table 2).

## 4. Discussion

With the effects of climate change becoming constantly more frequent and severe, policymakers and scientists are working strenuously to find solutions to mitigate the most catastrophic consequences. In this scenario, it is of pivotal importance to monitor and investigate all possible impacts that these solutions themselves might have on the ecosystems they are supposed to protect.

Although the MOSE might help preserving the city from future flood events, recent studies claimed that it would only be effective for moderate sea level rises, that repeated and prolonged closures will be necessary [7] and that this would rapidly deplete oxygen level in the lagoon [7,15,16], harming the natural populations of macro- and microorganisms, and causing a non-negligible modification of the entire lagoonal asset [78]. The MOSE closures exert numerous potential impacts on the lagoon: (i) hydrodynamic reduction, (ii) changes in water physical-chemical characteristics, (iii) impairment of the tidal regime, (iv) reduction/increase of sea-lagoonal sediment exchange, (v) alteration of the life cycles of organisms and, overall, of lagoon communities and ecosystem functioning.

Our data confirm a non-negligible impact of the MOSE closure on the MPB community. In particular, the specific community composition significantly changed from the beginning through the end of both the experiments, and even more evidently in summer compared to autumn. In general, the reduced hydrodynamics and the isolation of the lagoon environment from the open sea, caused by the rise of the MOSE gates, limit oxygen and nutrients exchanges and favor the gradual sinking of the particulate matter from the water column toward the sediment.

### 4.1. Microphytobenthic Community through Classical Taxonomy

From a quantitative point of view, the MPB abundance, particularly of diatoms, doubled from the beginning to the end of the experiments (Figure 2). This rapid increase was likely ascribable to TOC enrichment in the surface sediments (Figure 4), and the consequent release of inorganic nutrients, associated with the deposition of fresh organic material (lipids and proteins, Table 1). Indeed, in the BIO-ENV analysis, TOC consistently emerged as one of the best-correlated variables in both experiments. This is in accordance with previous studies reporting that the MPB community is strongly stimulated by high organic loads [79,80,81,82].

Overall, the structure of the benthic diatom communities observed during the two experiments was in line with those previously reported in the Venice Lagoon [18,83]. In particular, the high abundance of the planktonic genus *Thalassiosira* observed in July 2019 was in accordance with [83], who reported that *Thalassiosira* sp. represented ca. 90% of the total benthic diatom communities in the innermost part of the Venice Lagoon in August. Pelagic centric diatoms, once settled on the seafloor, become part of the MPB community when still intact and photosynthetically active [84]. This holds even more true in shallow environments such as lagoon systems.

The microalgal biodiversity inside the mesocosms varied during the experiment. Interestingly, in summer, from an initial state dominated by a few species, the community species richness increased (Figure 3), particularly on account of species belonging to the genera *Tryblionella* and *Psammodictyon* (Appendix A). *Nitzschia tryblionella* (currently a synonym of *Tryblionella hantzschiana*, www.algaebase.org, accessed on 23 March 2023) was described as a nutrient-loving species that thrives under high-organic matter conditions [85]. This species accounted for up to 19% of the MPB community in the Po Delta lagoon sediments that are rich in TOC contents derived from intensive clam farming [24]. Similarly, *Nitzschia panduriformis* (synonym of *Psammodictyon panduriforme*, www.algaebase.org, accessed on 23 March 2023) was also reported to be part of a phosphate-loving assemblage [86]. Further, in the LEfSe analysis, after *Psammodyction* cf. *constrictum* and *Tryblionella* cf. *hungarica*, *Nitzschia* cf *commutata* was the third most discriminating species between the beginning and the end of the summer experiment. This species was previously reported to have some affinity for TOC-enriched sediments since it reached up to 27% of the MPB relative abundance in a harbor area [82].

In contrast, other taxa, particularly *Paralia sulcata*, greatly reduced their abundance during the experiment. This is a tychopelagic diatom (i.e., a non-motile centric diatom) that lives loosely associated with sediments [87] and usually prefers low hydrodynamic conditions, as it would likely be washed away by currents at high bottom velocities [24]. Nevertheless, the lower hydrodynamics inside the mesocosms did not seem to favor this species. Other factors, such as a change in grain-size composition, with a slightly higher clay content following the deposition of fine suspended material, may have influenced the abundance of this species.

In October 2020, the total microalgal abundance increased but the biodiversity and species richness slightly declined toward the end of the experiment. Some taxa clearly benefited from the modified physical-chemical conditions inside the mesocosms, particularly those belonging to the genus *Gyrosigma*, namely *G. macrum* and *G. fasciola* (Figure 4 and Figure 5). *Gyrosigma* also prefers high organic content [88], and *G. macrum* and *G. fasciola* were reported as the most abundant species under a 20-year mussel farm in the Gulf of Trieste in September [81]. Similarly, *Mastogloia braunii* almost doubled its density from the beginning to the end of the experiment. This is a typical brackish and epipsammic genus, previously observed on sandy sediments of the Caleri Lagoon of the Po Delta [24] whose occurrence was found to be positively correlated with phosphate [89].

From the LEfSe analysis, *Surirella* emerged as the most discriminating taxa between the beginning and the end of the autumn experiment. We observed many highly silicified specimens in cell division (Cibic, personal comment), testifying that the induced lower hydrodynamics favored the reproduction of this taxon.

### 4.2. Microphytobenthic Community through Metabarcoding

In the sediments, together with typically benthic taxa, we detected some planktonic forms. This was observed especially in July 2019 and was due to cells sinking from the water column towards the sediments following the reduced hydrodynamics. This result was also an anticipated consequence derived from the reduced horizontal water fluxes in the mesocosms, similarly to what we could expect in case of MOSE closure. From both the indices and the LEfSe analysis, an overall loss of richness (d) and diversity (H’) in the sediments was evident, from the beginning toward the end of both the experiments, with many taxa disappearing or becoming significantly less abundant (21 and 44 in summer and in autumn, respectively), and few others appearing or becoming significantly more abundant (8 and 12 in summer and in autumn, respectively).

It is interesting to note that the taxa whose abundance significantly increased at T3 in October 2020 belonged almost entirely to the Naviculaceae family, which is known for its wide ecological valence. However, under various types of stress (e.g., metal pollution, high organic loads), tolerant taxa take over, and become dominant components of the MPB community [48,80,81].

As already evidenced in previous studies [32,33,34], by comparing the results of the microscopy counts with those of the metabarcoding, we found a good consistency between the two techniques in representing the taxonomic composition of the MPB communities in the two seasons. For instance, in summer 2019, the abundant planktonic genus *Thalassiosira* was identified by both techniques due to the size and the state of the still-intact cells in the surface sediments. In contrast, other genera, such as *Tryblionella* and *Psammodictyon,* were not detected through metabarcoding, almost certainly due to the fact that these genera are still considered part of the large genus *Nitzschia* (www.algaebase.org, accessed on 23 March 2023) in the used database. Similarly, the very abundant genus *Mastogloia* was likely listed as undetermined Pennales because the databases are not yet implemented with many pennate benthic diatom forms. In fact, it is possible that the benthic organisms are less characterized from a genomic point of view than the planktonic ones. Nevertheless, as a rule, the 18S sequencing was able to reach a much higher resolution within samples compared to classical taxonomy, and to detect many taxa non-detectable through microscopy. For instance, some diatoms were probably not identified by classical taxonomy due to their small size, e.g., the genus *Minutocellus* that belongs to the picophytoplankton (size < 3 um) [90], or the genus *Plagiostriata* [91]. Further, the molecular methodology detected many other taxa that were not visible under the microscope, and therefore could not be counted/identified, probably because they were ruptured. This is especially true for small phytoflagellates (i.e., Chrysophyceae, Prymnesiophyceae and Chlorophyta) without a hard theca or shell, and planktonic diatoms with a thin frustule, such as those belonging to the genus *Chaetoceros*. However, since 18S rRNA gene metabarcoding can also quantify DNA residues from dead organisms, e.g., it cannot discriminate between living photosynthetically active cells and dead organism/DNA residues, it leads to an overestimation of community biodiversity. In general, molecular tools may provide a more accurate picture of what is present in the community, but in some cases, and especially for larger cells such as diatoms, the classical taxonomy is more informative from a quantitative point of view, i.e., much more reliable in terms of the absolute abundances used in ecological studies. Moreover, we observed that databases on microalgae from surface sediments obtained from metabarcoding are still poorly implemented and often give an assignment only at a higher hierarchical level (e.g., family or order).

These results confirmed that both these techniques provide important but complementary information that should be integrated in studies aimed at investigating natural communities from both taxonomic and functional perspectives. In summary, molecular tools based on total DNA provide a picture of everything present in the community, dead or alive, while classical taxonomy can quantify what is alive at a given moment, which is of utmost importance in ecological studies.

### 4.3. Ecological Aspects of Altered MPB Composition

Pennate diatoms are capable of bio-stabilizing cohesive sediments against resuspension by secreting extracellular polymeric substances (EPS) [92] and are therefore crucial in erosion prevention of shallow water systems [93]. EPS production is species-specific; therefore, sediment stabilization varies depending on the composition of the MPB community [94,95]. For instance, *Navicula*, *Nitzschia*, *Amphora* and *Thalassiosira*, among others, are known to produce EPS, whereas *Paralia* is not an EPS-producer ([96] and references therein), and information on *Tryblionella* remains limited. Further, large-sized diatoms, e.g., *Nitzschia sigma* and *Surirella ovata,* have been reported to produce significantly high amounts of EPS because large, motile diatoms require greater EPS amounts for motility compared with small ones [94]. Recently, Kim and colleagues found that the ecological aspects of MPB, such as diatom composition and size, influence sediment biostabilization, and identified *Navicula* as the key diatom genus of this process, i.e., a “sediment stabilizer” [96]. Therefore, the modified composition of the MPB community induced by the MOSE closure could lead to potentially crucial changes in their ability to bio-stabilize cohesive sediments, if EPS-producers are largely replaced by non-EPS producers. Loosely bio-stabilized sediments are easily resuspended, which in turn, could have serious repercussions on water column turbidity [31] and overall primary production of the lagoon ecosystem.

## 5. Conclusions

To the best of our knowledge, this is the first study that investigated the effect of a storm-surge infrastructure by simulating in situ its impacts on biological communities through mesocosm experiments, by employing both classical taxonomy and molecular tools.

The MOSE is a monumental engineering project with enormous resources devoted to its realization. Its utility in attempting to protect Venice City from flooding deserves to be recognized. Nevertheless, Venice, its economy and its biological communities cannot be preserved unless the health and functioning of its lagoon is also preserved.

## Figures and Tables

**Figure 1 microorganisms-11-00936-f001:**
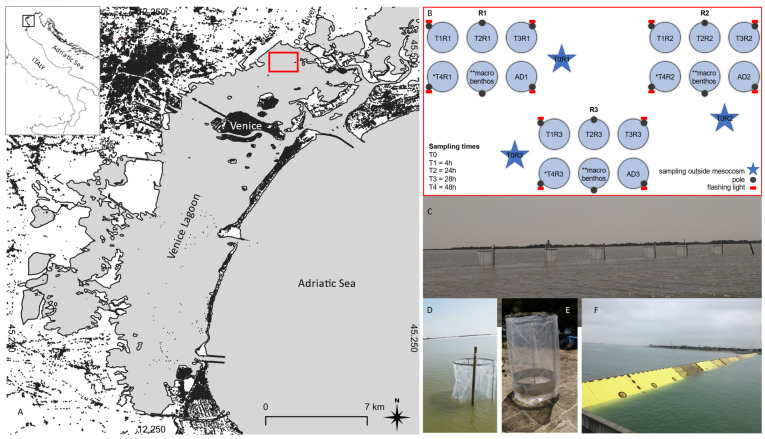
(**A**) Study area; (**B**) Experimental design: 18 mesocosms were placed in 3 groups of 6 to assure sampling in 3 replicates per 5 experimental times (* T4 not present in October 2020, ** not the topic of this work, see the text, ADn: additional backup mesocosms); (**C**) Mesocosms placed in Palude di Cona; (**D**,**E**) Detail of the mesocosms; (**F**) MOSE gates in operation (source: www.mosevenezia.eu, accessed on 23 March 2023).

**Figure 2 microorganisms-11-00936-f002:**
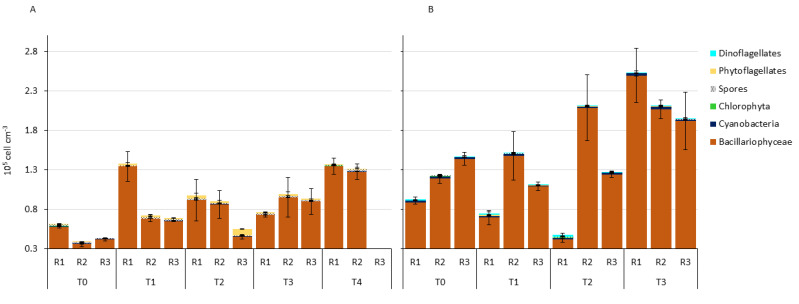
Microphytobenthic total abundance divided in the main microalgal groups for each sampling time (T) and enclosure replicate (R): (**A**) in July 2019 and (**B**) in October 2020.

**Figure 3 microorganisms-11-00936-f003:**
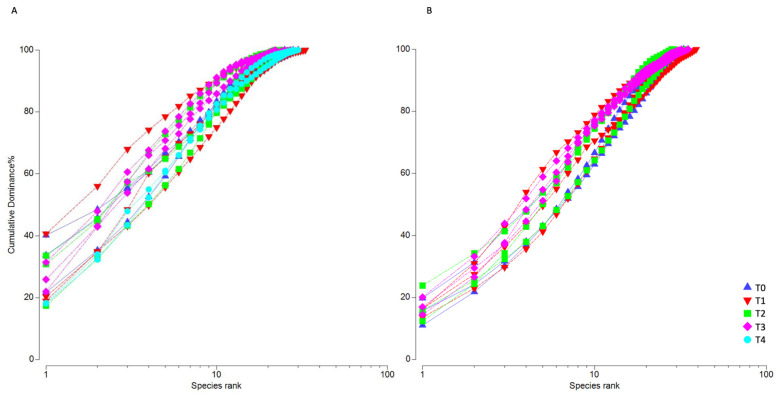
K-dominance curves applied to the microphytobenthic community divided by experimental time (n = 3) (**A**) in July 2019 and (**B**) in October 2020.

**Figure 4 microorganisms-11-00936-f004:**
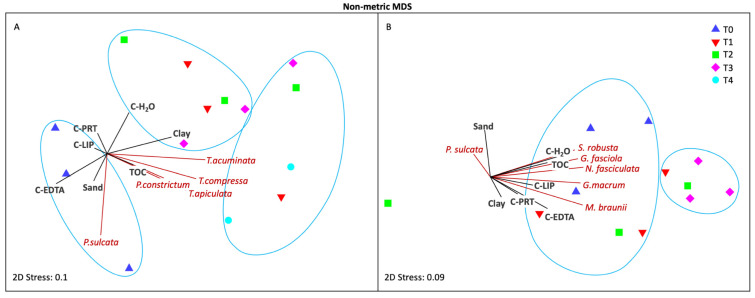
Non-metric MDS ordination plots calculated on the overall MPB abundances (**A**) from T0 to T4 in July 2019 and (**B**) from T0 to T3 in October 2020. The most abundant species (RA ≥ 5%) and the environmental variables are overlaid as vectors; circles delimit clusters obtained applying the SIMPROF analysis (complete linkage, similarity threshold = 0.05, n. of permutations = 999). (C-LIP = lipids, C-PRT = proteins, C-H_2_O = water soluble carbohydrates, C-EDTA = EDTA soluble carbohydrates).

**Figure 5 microorganisms-11-00936-f005:**
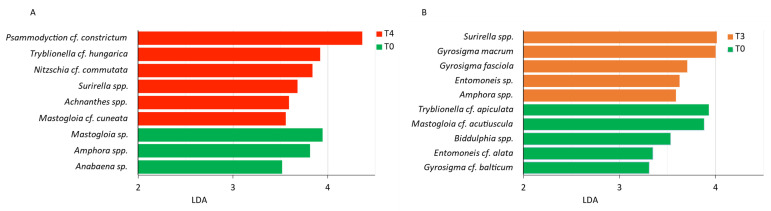
LEfSe results: MPB species discriminative with respect to the experimental time (α-value for the Kruskal–Wallis and Wilcoxon test = 0.05; logarithmic LDA score threshold = 2). (**A**) July 2019, (**B**) October 2020.

**Figure 6 microorganisms-11-00936-f006:**
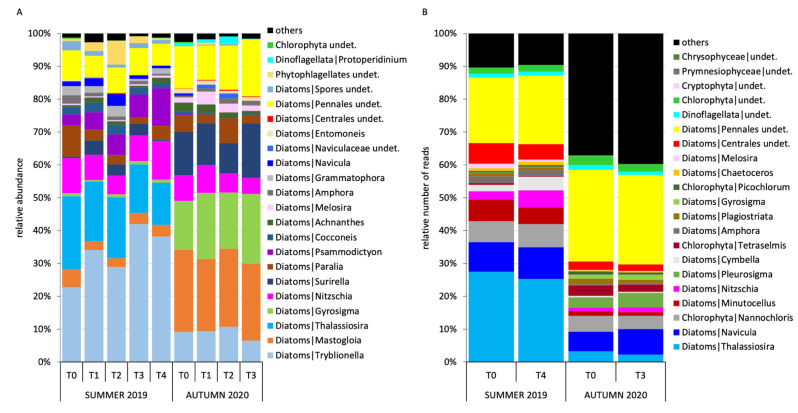
Comparison between the taxonomic composition of the MPB community obtained through (**A**) classical microscopy and (**B**) 18S RNA gene sequencing. The species/ASVs representing relative number of cells/number of reads ≥ 0.5% are shown; averages of three enclosure replicates at each sampling time are represented.

**Table 1 microorganisms-11-00936-t001:** BEST (BIO-ENV + STEPWISE) results: Groups of environmental variables that best correlate with the taxonomic composition of the MPB community, according to Spearman’s rank coefficient (n. of permutations = 999); (C-LIP = lipids, C-PRT = proteins).

	n. Variables	Correlation	Variables
**SUMMER 2019**	1	0.500	C-LIP
	2	0.375	TOC+C-LIP
	3	0.375	Sand+TOC+C-LIP
	4	0.500	Sand+Clay+TOC+C-LIP
	5	0.375	Sand+Silt+Clay+TN+C-LIP
**AUTUMN 2020**	1	0.625	C-LIP
	2	0.625	TN+C-PRT
	3	0.500	TN+C-PRT+C-LIP
	4	0.625	TN+TOC+C-PRT+C-LIP
	5	0.375	TN+Sand+Silt+Clay+C-PRT

**Table 2 microorganisms-11-00936-t002:** LEfSe results: MPB species, discriminative with respect to the experimental time (α-value for the Kruskal–Wallis and Wilcoxon test = 0.05; logarithmic LDA score threshold = 2). The taxonomic level that could be classified with the maximum likelihood (≥ 80%) is represented. (T0 = green; T4—Summer 2019 = red; T3—Autumn 2020 = orange).

A	SUMMER 2019		B	AUTUMN 2020	
	T0	LDA		T0	LDA
	Bacillariophyta.Araphidpennate.Fragilariales	3.35		Bacillariophyta.Staurosiraceae	3.57
	Chlorophyta.*Tetraselmis convolutae*	3.12		Bacillariophyta.Araphidpennate.Fragilariales endosymbiont	3.47
	Bacillariophyta.Araphidpennate.Fragilariales	2.98		Bacillariophyta.Raphidpennate.Naviculales	3.43
	Bacillariophyta.Araphidpennate.Fragilariales	2.98		Bacillariophyta.Araphidpennate.Fragilariales endosymbiont	3.41
	Bacillariophyta.Araphidpennate.Fragilariales	2.85		Dinoflagellata.*Gymnodinium smaydae*	3.30
	Bacillariophyta.Raphidpennate.*Amphora*	2.85		Bacillariophyta.Araphidpennate.Fragilariales	3.23
	Bacillariophyta.Araphidpennate.*Dimeregramma*	2.83		Bacillariophyta.Araphidpennate.Fragilariales	3.14
	Dinoflagellata.*Gymnodinium*	2.82		Chlorophyta.Nannochloris.*Picochlorum*	3.06
	Dinoflagellata.*Gymnodinium smaydae*	2.82		Bacillariophyta.Araphidpennate.Fragilariales endosymbiont	2.98
	Bacillariophyta.Raphidpennate.*Sellaphora*	2.75		Bacillariophyta.Araphidpennate.*Nanofrustulum shiloi*	2.85
	Dinoflagellata.*Luciella* sp.	2.75		Bacillariophyta.Raphidpennate.Naviculales	2.84
	Bacillariophyta.Naviculales	2.69		Chlorophyta.Picochlorum.*Nannochloris* sp. *MI37*	2.82
	Bacillariophyta.Araphidpennate.Fragilariales-Saurosira	2.66		Bacillariophyta.Araphidpennate.Fragilariales endosymbiont	2.82
	Bacillariophyta.Araphidpennate.Fragilariales	2.66		Chlorophyta.Picochlorum.*Nannochloris* sp. *MI37*	2.71
	Bacillariophyta.Polarcentric Mediophyceae.*Cyclotella striata*	2.61		Bacillariophyta.Araphidpennate.Fragilariales endosymbiont	2.66
	Bacillariophyta.Araphidpennate.*Plagiostriata goreensis*	2.52		Chlorophyta.*Picochlorum eukaryotum*	2.65
	Chlorophyta.*Desmodesmus communis*	2.51		Bacillariophyta.Araphidpennate.Fragilariales endosymbiont	2.65
	Bacillariophyta.Polarcentric Mediophyceae.*Skeletonema*	2.51		Chlorophyta.Chlorellales.*Nannochloris* sp. *MBIC10053*	2.64
	Cryptophyta.*Urgorri complanatus*	2.39		Chlorophyta.Chlorellales.Trebouxiophyceae	2.61
	Dinoflagellata.Clade4Xsp90	2.36		Dinoflagellata.*Blixaea quinquecornis*	2.57
	Bacillariophyta.Raphidpennate.Naviculales	2.27		Dinoflagellata.*Luciella* sp.	2.57
	**T4**	**LDA**		Bacillariophyta.*Pleurosigma* sp. *mgcode 4*	2.55
	Bacillariophyta.Raphidpennate.*Tryblionella*	3.40		Bacillariophyta.Naviculales	2.52
	Bacillariophyta.Raphidpennate.*Fragilariopsis sublineata*	2.93		Bacillariophyta.Araphidpennate.*Plagiostriata goreensis*	2.50
	Dinoflagellata.*Gymnodinium aureolum*	2.79		Bacillariophyta.Coscinodiscophyceae	2.48
	Chlorophyta.*Mantoniella antarctica*	2.70		Bacillariophyta.Polarcentric Mediophyceae.Thalassiosira.*Thalassiosira*	2.47
	Dinoflagellata.*Scrippsiella acuminata*	2.69		Chlorophyta.*Pseudoscourfieldia marina*	2.45
	Dinoflagellata.*Prorocentrum micans*	2.62		Chlorophyta.*Ostreococcus mediterraneus*	2.44
	Chlorophyta.*Chlamydomonas*	2.59		Chlorophyta.*Nannochloris* sp.	2.44
	Bacillariophyta.Raphidpennate.*Nitzschia amphibia*	2.47		Bacillariophyta.Polarcentric Mediophyceae.*Cyclotella meneghiniana*	2.44
				Dinoflagellata.Gymnodiniaceae	2.43
				Dinoflagellata.Symbiodiniaceae.*Symbiodinium*	2.41
				Bacillariophyta.Araphidpennate.Fragilariales endosymbiont	2.37
				Dinoflagellata.Thoracosphaeraceae	2.36
				Cryptophyta.Hemiselmis.*Hemiselmis cryptochromatica*	2.28
				Bacillariophyta.Polarcentric Mediophyceae.Odontellaceae	2.28
				Bacillariophyta.*Papiliocellulus elegans*	2.28
				Bacillariophyta.Polarcentric Mediophyceae.Chaetoceros.*Chaetoceros*	2.27
				Dinoflagellata.DinoGroupIClade1Xsp	2.25
				Dinoflagellata.*Heterocapsa niei*	2.23
				Bacillariophyta.Raphidpennate.*Craticula importuna*	2.22
				BacillariophytaRaphidpennate.Naviculales	2.20
				BacillariophytaRaphidpennate.*Pleurosigma sp. mgcode 4*	2.16
				Bacillariophyta.Araphidpennate.*Plagiostriata goreensis*	2.07
				**T3**	**LDA**
				BacillariophytaRaphidpennate.*Navicula*	4.10
				BacillariophytaRaphidpennate.Navicula.*Navicula*	3.53
				BacillariophytaRaphidpennate.Navicula.*Navicula*	3.26
				BacillariophytaRaphidpennate.*Pleurosigma*	3.23
				BacillariophytaRaphidpennate.*Navicula*	3.06
				BacillariophytaRaphidpennate.Navicula.*Navicula*	2.83
				Bacillariophyta.Raphidpennate.*Navicula cryptotenella*	2.75
				Bacillariophyta.Raphidpennate.*Navicula cryptotenella*	2.66
				Bacillariophyta.Raphidpennate.Naviculaceae	2.53
				Bacillariophyta.Raphidpennate.Naviculales	2.50
				Cryptophyta.*Hemiselmis tepida*	2.47
				BacillariophytaRaphidpennate.Naviculaceae	2.40

## Data Availability

Sequences are deposited at the Sequence Read Archive (SRA) under the project PRJNA915329.

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
