# Peer review of "The Impact of MOSE (Experimental Electromechanical Module) Flood Barriers on Microphytobenthic Community of the Venice Lagoon"

_microorganisms, 2023, doi:10.3390/microorganisms11040936_

Round 1

Reviewer 1 Report

Main concerns:

1.      The research background is overextension, and climate change seems to have little relevance to this study, although construction of MOSE was related to climate change.

2.      Why methodology was estimated? What aims were? If the aim was to evaluate reliability of method, it should be collected samples from different sources. In fact, the two methods have their own advantages and disadvantages. They can be combined to explain the scientific questions.

3.      Statistical analyses were not enough, for example whether there was statistically significant difference among all groups, as well as relationship between communities and environmental factors.

Minor concerns:

1.      The first appearance of MOSE should be given the full name.

2.      The pictures are incomplete

3.      The reference format is incorrect, for example, Cavaleri et al., 2020 and Umgiesser, 2020

4.      There are lack of the PCR reactions and reaction conditions.

5.      Some descriptions of statistical analyses were not incomplete, for example nMDS.

6.      nMDS is not best analysis for relationship between communities and environmental factors. Also, it is not proper to analyze at genus level.

7.      The descriptions about LEfSe were not accurate.

Author Response

Comments and Suggestions for Authors

Main concerns:

  1. The research background is overextension, and climate change seems to have little relevance to this study, although construction of MOSE was related to climate change.

We believe a thorough background is needed to introduce the topics covered in the paper and focus on some specific points. As pointed out by the Reviewer, the construction of MOSE is tightly related to climate change and therefore we had at least to refer to it. We had just briefly mentioned climate change at the beginning of the introduction and discussion sections. Further, Reviewer 2 suggested to implement the introduction with the most recent physical-chemical and biological data and add some comments on the future work to study the cumulative effect of the MOSE operational phase. Therefore, following the second Reviewer’s suggestions we have further extended this part.

  1. Why methodology was estimated? What aims were? If the aim was to evaluate reliability of method, it should be collected samples from different sources. In fact, the two methods have their own advantages and disadvantages. They can be combined to explain the scientific questions.

We are not sure to have understood the Reviewer’s comment. The aim of this study was not to evaluate the reliability of the method but to investigate the possible response of the Venice lagoon ecosystem, in terms of structure and function, following the closure of the MOSE. We had therefore designed an experimental design which is case-specific. To the best of our knowledge, there are no systems such as the MOSE around the world aiming at protecting a large, inhabited lagoon. We believe the aims of this work where clearly stated at the end of the introduction. In this manuscript we presented only a part of the results gathered during this experiment.

We emphasized the advantages and disadvantages of the two methods consistently with what already observed in previous studies (which we cited in the manuscript). In addition, we extended this knowledge by comparing these two techniques in an experimental setup in the context of a lagoonal system, as never, in our knowledge, has been done before.

  1. Statistical analyses were not enough, for example whether there was statistically significant difference among all groups, as well as relationship between communities and environmental factors.

Again, we are not sure to have understood the Reviewer’s comment. The statistically significant difference among the (samples) groups is showed by the SIMPROF clusters superimposed to the nMDS plots, we now clarified it better in the methods. The relation between communities and environmental factors is proven by the BEST (BIO-ENV + STEPWISE) analysis, as described in the methods section. We also previously constructed a Principal component analysis but since the focus of this paper is on the assemblages and not on physical-chemical results, we decided to not include this further analysis in this manuscript.

Minor concerns:

  1. The first appearance of MOSE should be given the full name.

Thank you for this suggestion, we added it in the title.

  1. The pictures are incomplete

Unfortunately, the Reviewer missed to indicate which pictures are incomplete and what kind of information is lacking. The other two Reviewers did not mention the incompleteness of the pictures, one of them only suggested to enlarge the characters in Figure 1B, which we have now done.

  1. The reference format is incorrect, for example, Cavaleri et al., 2020 and Umgiesser, 2020

We checked and corrected the reference format.

  1. There are lack of the PCR reactions and reaction conditions.

We now added the PCR reaction details.

  1. Some descriptions of statistical analyses were not incomplete, for example nMDS.

We now added the statistical method we used to cluster the samples groups.

  1. nMDS is not best analysis for relationship between communities and environmental factors. Also, it is not proper to analyze at genus level.

We constructed the nMDS analysis on the biological assemblages only and then plotted the environmental factors as additional variables, meaning that the environmental variables were not included in the analysis, but only graphically added in a second moment to visualize the interaction between biotic and abiotic factors. We did not apply this analysis on the genus level but on the species level. Indeed, on the two nMDS plots (Figure 4) the species are superimposed (first letter = genus, followed by the species name).

  1. The descriptions about LEfSe were not accurate.

We don’t understand what the reviewer requires, the LEfSe description is taken directly from its official reference.

Reviewer 2 Report

The manuscript refers to the impact the MOSE flood barriers on the microphytobenthos of the Venice lagoon, Italy. When the barriers are operational, the hydrodynamics is reduced, altering the water renewal, sediment fluxes and oxygen availability, modifying the lagoon and impacting the ecosystem. Venice, an important area for its socio-economic interest and ecosystem preservation, was suffering flooding due to extreme high tides, predicted to be even worse and more frequent with advancing climate change. As the barriers are already installed and working for a couple of years, the early study of the physicochemical and ecological alterations of the lagoon environment is relevant not only for scientists but also for politicians and the whole community, to understand and preserve the ecosystem, the climate, and the world.

General comments:

The manuscript is presented in a well-structured manner and the cited references are relevant and include recent publications. The experimental design is rather appropriate to test the hypothesis, that is sound and clear. But, to my appreciation, it should mention that the modifications in MPB abundance refer to total abundance, as it can be interpreted as relative abundances used to describe the community structure. In the following sentence it is mentioned that the higher input of particulate matter is predicted to promote a higher absolute abundance of MPB, but it’d be better to mention it in the hypothesis. Also, as hypothesis are on an idea or proposition that can be tested by observations or experiments and then supported or refused, they should be written in the present tense. Therefore, instead of “might induce” I consider it’d be proper to write “will induce” or “are inducing”.

Specific comments:

Introduction

The MOSE system started to operate about two years ago, so it would be good to have a brief comment in the introduction in addition to the prediction for the near future, also on the progression of the actual physicochemical/biological alterations detected or reported to the present. Also, if the monitoring of the change in observations made in this study will be repeated in the future at some time intervals to see the cumulative effect of the MOSE operations, done by you or other research or governmental institution.

At the end (from Furthermore…) an additional aim of testing which of the two methodologies used provides the most reliable representation of the MPB community structure followed by a more detailed explanation of the expected outcome is mentioned. What do you refer to as “the most reliable representation”? compared to what to judge the results as reliable? Moreover, a couple of paragraphs before it is referred to the use of both as Integrated Taxonomy, mentioning that the two approaches (molecular and classical taxonomy) are complementary and provide different but equally important information. Integrated (or Integrative) taxonomy aims to describe the diversity of living organisms from multiple and complementary perspectives, not only ribosomal DNA sequences and microscopy observations but also phylogeography, population genetics, ecology, physiology, etc. To test which approach is more reliable to represent the community structure or to use them as complementary is not the same. Could this be expressed more clearly? Maybe not mention the integrative taxonomy if the study involves mostly single-celled photosynthetic organisms and only DNA sequencing and microscopy will be used? And is classical taxonomy considered here to be equivalent to operator-driven classical microscopy? And more reliable for quantification of living taxa? The molecular approach can also be done from RNA extraction and cDNA sequencing to detect actively living taxa.

Materials and methods

It is mentioned that the 18 mesocosms were situated along transects with similar depth (< 75 cm), but how far apart were one mesocosm from the next one in one row and how distant were the rows between them?

Table S1 is cited here, but it has the results measured. It should be included in the Results section and commented if the differences, if observed, are statistically significant or not and the ecological implicancies. At T0, the measures were also done inside and outside the mesocosms? because the values are equal even at the decimal level. If, as it was expected that before the closure of the mesocosms there will not be differences in the parameters, if measurement was done from only one side it would be mentioned in the text and in the legend of Table S1.

The long-term sampling times T2 and T4 were after 24 h and 48 h of the induced closure of the mesocosm. Why T3 was after 28 h and not 36 h? And it is reliable or significant the information of T4 if there was one replicate lost and samples could not be taken in autumn? How long, in average, are usually the times the MOSE are kept operating and closed?

At the end of Experimental design, the names of the chemical analyses are written with capital letters, but then are defined the acronyms in Physical and chemical analyses and only carbon is in capital letter. Try to define the acronyms the first time that is mentioned (with capital letters or not) and then write only the acronym without repeating the full name.

In Abundance and community structure of MPB using classical taxonomy, the term MPB refers to the eukaryotic microalgae and the photosynthetic prokaryotes. Only the microscopy used to count all the photosynthetic specimens and describe the morphology of microalgae and the corresponding taxonomy keys are mentioned. Which classical taxonomy approach and taxonomy references were used to identify and classify the photosynthetic prokaryotes to have the community structure of the referred to as MPB?

In the taxonomy approach used to study community composition through metabarcoding, photosynthetic prokaryotes were not included as done with the other approach and where they were referred to as part of the MPB? Apart of the primers useful to target microalgae there are other primers to target prokaryotes. If one approach, as used, did not detect one group of organisms, both cannot be compared between them to see which is more reliable to evaluate community structure. If the molecular approach was used only to detect the eukaryotic part of the MPB, the classical taxonomy approach should be limited to describe the same part and the manuscript should be focused not on the study of MPB but on the study of microalgae response to the modifications caused by the operation of MOSE in the Venice Lagoon.

You mention the removal of 18S sequences assigned to chloroplasts, mitochondria, bacteria and archaea? did you have those sequences? It sound strange because those organisms do not have 18S rRNA, but if so, it was correct to remove them.

In Statistical analysis, the two biotic matrices mentioned, can be included as supplementary material and cited here when are mentioned for the first time? Or at least only one complete table with all the taxonomic rank and would be enough with the mention to which one each analysis was done. What do you mean with “all taxa at the genus and species levels”? that not all detected taxa could be identified to the species level?

With “univariate diversity analysis”, do you mean alpha diversity (within-sample diversity) and with “multivariate analysis”, beta diversity (between-samples diversity comparison)?

Results

The differences detected at most sampling times in the total microphytobenthic abundance estimated by microscopy for the three replicates, is expected? There are other studies showing that divergence? Did you performed any statistical analysis to find out if the dispersion between sampling times is significantly higher than the dispersion between replicates in a single sampling time (i.e. T1 or T2)?

 You mentioned to have identified three Cyanobacteria genera with this approach. How can you be sure there was no other genera displaying similar morphologies? Microscopy is not an acceptable method to reliable identify prokaryotes, not even filamentous Cyanobacteria. Several physiological and genetic traits are also employed in the classical taxonomy of prokaryotes.

 Also, when you mention the relatively low abundances of the Cyanobacteria genera identified (or the rather thin corresponding bars in Figure 2), which was the strategy used to count individuals? Cyanobacteria are rather smaller than microalgae and are unicellular, but use to organize in colonies of attached individuals, making easy to recognize them but almost impossible to count individual cells to calculate the abundance by microscopy. In my opinion, if cyanobacteria will be included as component of the MPB, they should be enumerated by qPCR using specific primers or other methodology to have a reliable abundance (and include them as well in the other approach sequencing 16S instead of 18S rDNA). Otherwise, not to include prokaryotes and focus the work on microalgae.

 When describing the k-dominance curves, the sentence “This result was in agreement with the distribution of species represented in the n-MDS ordination” refers to something that at that point was not yet shown. Better move the comment to where the ordination is presented. Note that here n-MDS was written with a dash and the other times nMDS was written without the dash.

If possible, place the nMDS results of the two experiments together and then the LEfSe and Bioenv results, with their corresponding comments.

 About the sentence “In October 2020, two groups emerged in the nMDS ordination: one constituted by all T0 samples, besides other samples, and the other comprising all T3 samples (Fig. 4B).”, I also see other samples in the second group marked on the plot 4B, one replicate from T1 and one from T2, together with the three T3 replicates.

The groups displayed with the ellipses in the nMDS plot, were defined by the analysis and have statistical support? To me, looking at plain sight, it seems as the second group would be integrated by the two T4 samples plus one T1 replicate, influenced by the TOC and four taxa, and the third group would include the other two T1 replicates plus the three T2 and T3 samples, influenced mostly by clay.

How are the results of the fit to nMDS ordination (vector length, significance) and Bioenv jointly interpreted to predict the influence of environmental parameters on microalgal community composition?

You mention that “The rarefaction curves for all the samples had reached a plateau, despite the samples having a different sequencing depth”, did you take the same number of sequences in all samples for uni and multivariate analysis to compare between the samples? Please, include the rarefaction curves in a single plot as supplementary information.

What do you consider to be the explanation for the fact that half of the ASVs could not be classified to the higher taxonomic rank (“The initial number of total ASVs was 10288; of these 4980 were not assigned to any phylum…”)?

In the Figure 6 legend it is mentioned that “averages of three enclosure replicates at each sampling time are represented”. How big were the differences in the number of sequences between the three replicates at each sampling time?

 Why the LEfSe results from the two taxonomic identification approaches (microscopy and metabarcoding) were represented in a different way (Figure 5 and Table 2)?

 Figure 1: Image A is clear but the images B to F are too small to appreciate the details or read the text, even printed, making necessary to amplify them with the computer.  Also, the distribution of the mesocosms in image B is confusing if it intends to plot the experiment design in situ as three rows like the one showed in the image C. Moreover, all the references inside the circles of R1, R2 and R3 groups say R1. If it were not possible to make two separate figures or transfer some images to the supplementary information and display them in a bigger size, the panels could be organized differently to better appreciate what the images show, what it is useful to understand the explained in the text (i.e. A smaller, on its right D and F and below the two F, longer, with less sky but showing clearly the MOSE up, and below the four images, B with the mesocosms as three rows of six and below C, aligned to B but showing a real image of a row of mesocosms). In the reference, where it says “** not the topic of this work”, could it be included “see the text” to make it more understandable? And what is referred to as AD1?

Figure 3: What do you mean by “grouping the replicates of each experimental time”?

 Figure 4: All the fitted taxa and physicochemical parameters vectors had a significant correlation with the nMDS ordination? Is the origin of all plotted vectors in the 0,0 centered axes of the ordination of the samples? If so, in 4B it would appear to be a single group and not two. The view of both plots should show the 0,0 intersection of the NMDS1 and NMDS2 axes, including the scaling values and the labels.

Table S2: The taxonomic identification of the taxa included in this table was done by classical taxonomy or by sequencing? the approach used should be mentioned in the legend of the table.

 Table S3: The legend starts "Diversity indices applied to MPB abundances in the two experiments" but I consider it'd be more appropriate "Diversity indices of MPB communities sampled in the two experiments"

Tables S3 and S4: Were the indices calculated with about the same number of reads per sample? If the number of reads/individuals per sample varies by more than about 10%, the calculated parameters could not be comparable to draw conclusions.

 Discussion

You refer to recent studies that claimed that the closure of MOSEs would only be effective for moderate sea level rises and that repeated and prolonged closures will be necessary and rapidly deplete oxygen level in the lagoon. Was that reducing in oxygen level observed/confirmed with these experiments? Or there are evidence published by other authors of the reduction in O2 concentrations since the implementation of MOSEs?

To “…limit oxygen and nutrients exchanges and favor the gradual sinking of the particulate matter from the water column toward the sediment (data not published)”, there are some general results now to include in this manuscript or any personal communication to cite?

 You mention that “This rapid increase was likely ascribable to TOC and TN enrichment in the surface sediments (Fig. 4)…” but TN is not included in the Fig.4 and there is no explanation about that. In the October experiment, in that figure the increase seems to be also related to BPC but in July experiment it seems to be the contrary.

To “In contrast, metabarcoding can also quantify DNA residues from dead organisms…”, it can be avoided by working with cDNA obtained from extracted total RNA, instead of extracting total DNA. That way, only the active organisms synthesizing RNA will be detected, because RNA is less stable than DNA and no residues from dead cells will be recovered.

Please, reformulate the last part of the 4.2 metabarcoding section taking into account all the above commented. The last sentence should be modified to convey the idea that molecular tools based on the extraction of total DNA provide a more accurate picture of what is present in the community…

 Instead of “Ecological valence of altered…” in the 4.3 section title “Ecological value of altered…” would be better.

Author Response

Comments and Suggestions for Authors

The manuscript refers to the impact the MOSE flood barriers on the microphytobenthos of the Venice lagoon, Italy. When the barriers are operational, the hydrodynamics is reduced, altering the water renewal, sediment fluxes and oxygen availability, modifying the lagoon and impacting the ecosystem. Venice, an important area for its socio-economic interest and ecosystem preservation, was suffering flooding due to extreme high tides, predicted to be even worse and more frequent with advancing climate change. As the barriers are already installed and working for a couple of years, the early study of the physicochemical and ecological alterations of the lagoon environment is relevant not only for scientists but also for politicians and the whole community, to understand and preserve the ecosystem, the climate, and the world.

We thank the reviewer for her/his suggestions and comments,

We below addressed the presented points.

General comments:

The manuscript is presented in a well-structured manner and the cited references are relevant and include recent publications. The experimental design is rather appropriate to test the hypothesis, that is sound and clear. But, to my appreciation, it should mention that the modifications in MPB abundance refer to total abundance, as it can be interpreted as relative abundances used to describe the community structure.

We have better explained that the modifications concern both the MPB total abundance and the community composition.

In the following sentence it is mentioned that the higher input of particulate matter is predicted to promote a higher absolute abundance of MPB, but it’d be better to mention it in the hypothesis. Also, as hypothesis are on an idea or proposition that can be tested by observations or experiments and then supported or refused, they should be written in the present tense. Therefore, instead of “might induce” I consider it’d be proper to write “will induce” or “are inducing”.

We have rewritten the hypotheses following the Reviewer’s suggestions.

Specific comments:

Introduction

The MOSE system started to operate about two years ago, so it would be good to have a brief comment in the introduction in addition to the prediction for the near future, also on the progression of the actual physicochemical/biological alterations detected or reported to the present. Also, if the monitoring of the change in observations made in this study will be repeated in the future at some time intervals to see the cumulative effect of the MOSE operations, done by you or other research or governmental institution.

The MOSE has become fully operational only 2 years ago, therefore there are not many data on current physicochemical/biological alterations yet. We have added a sentence from the most recent available literature on this topic.

It is not likely that our research group (or others, at the best of our knowledge) is going to repeat these observations since at the moment there is no funding dedicated to this kind of monitoring.

At the end (from Furthermore…) an additional aim of testing which of the two methodologies used provides the most reliable representation of the MPB community structure followed by a more detailed explanation of the expected outcome is mentioned. What do you refer to as “the most reliable representation”? compared to what to judge the results as reliable?

We have rewritten this sentence to make it clearer.

Moreover, a couple of paragraphs before it is referred to the use of both as Integrated Taxonomy, mentioning that the two approaches (molecular and classical taxonomy) are complementary and provide different but equally important information. Integrated (or Integrative) taxonomy aims to describe the diversity of living organisms from multiple and complementary perspectives, not only ribosomal DNA sequences and microscopy observations but also phylogeography, population genetics, ecology, physiology, etc. To test which approach is more reliable to represent the community structure or to use them as complementary is not the same. Could this be expressed more clearly? Maybe not mention the integrative taxonomy if the study involves mostly single-celled photosynthetic organisms and only DNA sequencing and microscopy will be used? And is classical taxonomy considered here to be equivalent to operator-driven classical microscopy? And more reliable for quantification of living taxa? The molecular approach can also be done from RNA extraction and cDNA sequencing to detect actively living taxa.

Following the Reviewer’s suggestion, we have removed the term Integrated taxonomy. Yes, in this ms we considered classical taxonomy as a synonym of operator-driven classical microscopy. We have better explained this term in the text. We are aware that the molecular approach can also be done from RNA extraction to detect living taxa, but in this kind of studies the 16S and 18S rRNA genes sequencing are more commonly used, therefore we applied them in our experiments. The classical microscopy, at least for the identification of larger diatom forms, was in our opinion more reliable than data obtained from 18S sequencing. In the next future we will rather opt for RNA extraction.

Materials and methods

It is mentioned that the 18 mesocosms were situated along transects with similar depth (< 75 cm), but how far apart were one mesocosm from the next one in one row and how distant were the rows between them?

We have added this information in the text.

Table S1 is cited here, but it has the results measured. It should be included in the Results section and commented if the differences, if observed, are statistically significant or not and the ecological implicancies.

We have moved Table S1 (now S3) in the results and have briefly commented it.

At T0, the measures were also done inside and outside the mesocosms? because the values are equal even at the decimal level. If, as it was expected that before the closure of the mesocosms there will not be differences in the parameters, if measurement was done from only one side it would be mentioned in the text and in the legend of Table S1.

No, at T0 there was not really an inside and outside since at T0 the nylon was just pulled up and there were no differences, only the outside was measured. We have added this piece of information in the legend of Table S1 (now S3) and in the text.

The long-term sampling times T2 and T4 were after 24 h and 48 h of the induced closure of the mesocosm. Why T3 was after 28 h and not 36 h? And it is reliable or significant the information of T4 if there was one replicate lost and samples could not be taken in autumn? How long, in average, are usually the times the MOSE are kept operating and closed?

We started the experiment at 9.00 a.m. on day 0. T1 was performed after 4 h (1.00 pm) the same day, T2 at 9.00 on day +1 (24 h) and T3 at 1.00 pm on day +1 (28 h). T3 could not be performed after 36h since it would mean starting sampling in the field at 7 p.m. The boats of the CNR ISMAR are not allowed to drive at night, we were in the middle of the lagoon, far from the urban settlements. We believe the information at T4 was still reliable also with two replicates only. These replicates were taken from two mesocosms with a sediment area of 0.785 m each, quite a representative portion of the lagoon. From each of the sampled sediment replicates, three counts were performed. In autumn the environmental conditions differed greatly from the summer ones, and the two obtained datasets were not comparable, in fact they were treated separately. On 4-6 December 2020, the MOSE system was closed for 48 hours nonstop, and this was the longest closure considering the closures performed till then. Usually the closures are much shorter (from a few hours to 24 hours) and the MOSE is activated only in very high tide conditions (105 cm), this can occur up to 4-5 times in each autumn month.

At the end of Experimental design, the names of the chemical analyses are written with capital letters, but then are defined the acronyms in Physical and chemical analyses and only carbon is in capital letter. Try to define the acronyms the first time that is mentioned (with capital letters or not) and then write only the acronym without repeating the full name.

We corrected accordingly.

In Abundance and community structure of MPB using classical taxonomy, the term MPB refers to the eukaryotic microalgae and the photosynthetic prokaryotes. Only the microscopy used to count all the photosynthetic specimens and describe the morphology of microalgae and the corresponding taxonomy keys are mentioned. Which classical taxonomy approach and taxonomy references were used to identify and classify the photosynthetic prokaryotes to have the community structure of the referred to as MPB?

The photosynthetic prokaryotes were identified and classified using the taxonomic key by Lund & Lund (Freshwater Algae - Their microscopic world explored) already cited among the taxonomic keys.

In the taxonomy approach used to study community composition through metabarcoding, photosynthetic prokaryotes were not included as done with the other approach and where they were referred to as part of the MPB? Apart of the primers useful to target microalgae there are other primers to target prokaryotes. If one approach, as used, did not detect one group of organisms, both cannot be compared between them to see which is more reliable to evaluate community structure. If the molecular approach was used only to detect the eukaryotic part of the MPB, the classical taxonomy approach should be limited to describe the same part and the manuscript should be focused not on the study of MPB but on the study of microalgae response to the modifications caused by the operation of MOSE in the Venice Lagoon.

The microphytobenthos include the microscopic aquatic organisms living on or close to the bottom. Most are eukaryotic (diatoms, dinoflagellates, phytoflagellates, etc.), but some prokaryotic photosynthetic organisms, such as cyanobacteria, also contribute to the MPB community (MacIntyre et al., 1996). In this study we included all the organisms that are part of the MPB to study the response of the entire community to the modifications caused by the operation of MOSE in the Venice Lagoon. Since cyanobacteria identified using classical microscopy were very sporadic (on average 0.67% in the first and 1.5% in the second experiment) we do not believe that excluding them would lead to much different results. Instead, it would not be consistent with previous studies that included all the components of the MPB and we believe this would not be worth doing since the comparison between the two methodologies (classic microscopy vs molecular tools) was only one of the aims of this study. However, we are aware of this methodological shortcoming, therefore we added a sentence in the introduction, to explain that cyanobacteria were not included in this comparison since in this work we applied primers to target eukaryotes only.

You mention the removal of 18S sequences assigned to chloroplasts, mitochondria, bacteria and archaea? did you have those sequences? It sound strange because those organisms do not have 18S rRNA, but if so, it was correct to remove them.

The reviewer is right. Within this huge experimental framework, we collected and processed also data from the 16S rRNA gene sequencing; this sentence is just a misplaced description of those data we are not presenting here; we removed it from the text.

In Statistical analysis, the two biotic matrices mentioned, can be included as supplementary material and cited here when are mentioned for the first time? Or at least only one complete table with all the taxonomic rank and would be enough with the mention to which one each analysis was done. What do you mean with “all taxa at the genus and species levels”? that not all detected taxa could be identified to the species level?

We now provided the two taxa tables as supplementary material.

Precisely, not all taxa could be identified to the species level, therefore we left them at the genus level.

With “univariate diversity analysis”, do you mean alpha diversity (within-sample diversity) and with “multivariate analysis”, beta diversity (between-samples diversity comparison)?

“Univariate" refers to the K-dominance and diversity indices analyses, “multivariate” refers to nMDS and BIO-ENV, we now changed that part in the methods section and hopefully made it clearer.

Results

The differences detected at most sampling times in the total microphytobenthic abundance estimated by microscopy for the three replicates, is expected? There are other studies showing that divergence? Did you performed any statistical analysis to find out if the dispersion between sampling times is significantly higher than the dispersion between replicates in a single sampling time (i.e. T1 or T2)?

If the Reviewer refers to the standard deviation of the three samplings, yes, this is in line with many other studies. These are not analytical replicates but biological replicates. Sediments were sampled in mesocosms that were about 100 m apart. This is a huge difference for such a small community as the microphytobenthos. Most estimates of the biological communities’ abundance (not only of benthic microalgae) display a SD of 20-25%. The occurrence of taxa is generally very patchy: when diatom colonies are present, the cell numbers may remarkably differ, and this leads to high SD considering the three biological replicates.

You mentioned to have identified three Cyanobacteria genera with this approach. How can you be sure there was no other genera displaying similar morphologies? Microscopy is not an acceptable method to reliable identify prokaryotes, not even filamentous Cyanobacteria. Several physiological and genetic traits are also employed in the classical taxonomy of prokaryotes.

In this case we identified three very easily recognizable cyanobacteria genera such as Spirulina, Oscillatoria and Anabaena. There are no other genera displaying similar morphologies. We used the taxonomic key of Lund &Lund: there is a chapter of more than 40 pages with illustrations and detailed descriptions dedicated to the identification of cyanobacteria. There are many other taxonomic keys available in the literature for their identification.

 Also, when you mention the relatively low abundances of the Cyanobacteria genera identified (or the rather thin corresponding bars in Figure 2), which was the strategy used to count individuals? Cyanobacteria are rather smaller than microalgae and are unicellular, but use to organize in colonies of attached individuals, making easy to recognize them but almost impossible to count individual cells to calculate the abundance by microscopy. In my opinion, if cyanobacteria will be included as component of the MPB, they should be enumerated by qPCR using specific primers or other methodology to have a reliable abundance (and include them as well in the other approach sequencing 16S instead of 18S rDNA). Otherwise, not to include prokaryotes and focus the work on microalgae.

The single cells of colonial cyanobacteria may be very large e.g. those of the huge Oscillatoria, or small e.g. in Anabaena. Although we generally used a 320x or 400x magnification, in case of doubt we were able to switch to a higher magnification, up to 640x to count the individual cells. This is not impossible, and it is performed routinely when counting and identifying the very small phytoplankton cells in monitoring programs carried out all around the world by environmental agencies, both in freshwater and coastal environments. In our case, the individuals we encountered were very few, 1-2 colony in each replicate, leading to very low overall cyanobacteria abundance. We added a sentence to specify that cyanobacteria could not be included in the comparison between the two methods.

 When describing the k-dominance curves, the sentence “This result was in agreement with the distribution of species represented in the n-MDS ordination” refers to something that at that point was not yet shown. Better move the comment to where the ordination is presented. Note that here n-MDS was written with a dash and the other times nMDS was written without the dash.

We edited accordingly.

If possible, place the nMDS results of the two experiments together and then the LEfSe and Bioenv results, with their corresponding comments.

The results of all these statistical tools contribute together to the characterization of each of the two experiments; therefore, we think it is more informative to divide the results according to experiment instead that according to statistical analysis.

 About the sentence “In October 2020, two groups emerged in the nMDS ordination: one constituted by all T0 samples, besides other samples, and the other comprising all T3 samples (Fig. 4B).”, I also see other samples in the second group marked on the plot 4B, one replicate from T1 and one from T2, together with the three T3 replicates.

We clarified this better.

The groups displayed with the ellipses in the nMDS plot, were defined by the analysis and have statistical support? To me, looking at plain sight, it seems as the second group would be integrated by the two T4 samples plus one T1 replicate, influenced by the TOC and four taxa, and the third group would include the other two T1 replicates plus the three T2 and T3 samples, influenced mostly by clay.

The groups were defined by a statistical test (SIMPROF), we forgot to mention this in the methods, now we added this piece of info in the text and in the figure caption.

How are the results of the fit to nMDS ordination (vector length, significance) and Bioenv jointly interpreted to predict the influence of environmental parameters on microalgal community composition?

The two analyses provide different information and we included different variables therefore it is not possible to jointly interpret these results.

You mention that “The rarefaction curves for all the samples had reached a plateau, despite the samples having a different sequencing depth”, did you take the same number of sequences in all samples for uni and multivariate analysis to compare between the samples? Please, include the rarefaction curves in a single plot as supplementary information.

As requested by the reviewer, rarefaction curves are now added in a single plot as Figure S1.

The only statistical analyses performed on the metabarcoding data are the diversity indexes; as we clarified below, we now normalized the ASV table and rerun the analyses.

What do you consider to be the explanation for the fact that half of the ASVs could not be classified to the higher taxonomic rank (“The initial number of total ASVs was 10288; of these 4980 were not assigned to any phylum…”)?

We thank the reviewer for the proper question, which indeed does not have a single, definitive answer. The PR2 database was chosen because it focuses mainly on unicellular eukaryotes, and it may not be particularly curated for certain taxa of multicellular organisms. At the same time, it is possible that the organisms in the sediment are less characterized from a genomic point of view than the planktonic ones.

We further studied this “not classified ASVs”, which account for 48.4% of the initial ASVs. On average these ASVs accounted for 13 ± 5.87% of the total counts in the samples. 781 were actually assigned to the Supergroup level of the PR2 database, divided as follows: Alveolata (115), Amoebozoa (125), Archaeplastida (43), Excavata (5), Hacrobia (4), Opisthokonta (246), Rhizaria (17), Stramenopiles (226).

In the Figure 6 legend it is mentioned that “averages of three enclosure replicates at each sampling time are represented”. How big were the differences in the number of sequences between the three replicates at each sampling time?

What is represented in Fig.6 are relative abundances, therefore the absolute n. of sequences/sample is not taken into account. As mentioned above, these are biological replicates and the variability is within the range common for this kind of studies. For the metabarcoding data, the n. of sequences/sample varied between 13 and 16% in the two experimental times of the first experiment and between 16 and 26% in the two experimental times of the second experiment.

 Why the LEfSe results from the two taxonomic identification approaches (microscopy and metabarcoding) were represented in a different way (Figure 5 and Table 2)?

Because, since the taxa table of the metabarcoding is much bigger than that obtained through classical taxonomy, it was not possible to represent it as a bar-chart.

 Figure 1: Image A is clear but the images B to F are too small to appreciate the details or read the text, even printed, making necessary to amplify them with the computer.  Also, the distribution of the mesocosms in image B is confusing if it intends to plot the experiment design in situ as three rows like the one showed in the image C. Moreover, all the references inside the circles of R1, R2 and R3 groups say R1. If it were not possible to make two separate figures or transfer some images to the supplementary information and display them in a bigger size, the panels could be organized differently to better appreciate what the images show, what it is useful to understand the explained in the text (i.e. A smaller, on its right D and F and below the two F, longer, with less sky but showing clearly the MOSE up, and below the four images, B with the mesocosms as three rows of six and below C, aligned to B but showing a real image of a row of mesocosms). In the reference, where it says “** not the topic of this work”, could it be included “see the text” to make it more understandable? And what is referred to as AD1?

The reviewer is right, Fig. 1B is a general schematic representation of both the experimental setups; not necessarily faithful to what was realized in the end due to logistic constrains.

We corrected as much as possible the figure and its legend, without splitting it in two separate figures, since we already have many supplementary tables and figures. This was only a pdf constructed to facilitate the review process; however, we prepared and uploaded a much larger figure in high resolution in which the smaller figures were very clear and easily readable. In the editing phase, we will ask to use half a page to print this figure.

We corrected the “R1” repetition and included a description for “AD” in the caption.

Figure 3: What do you mean by “grouping the replicates of each experimental time”?

We meant that each curve represents one experimental time, we now changed the figure caption and hopefully, made it clearer.

 Figure 4: All the fitted taxa and physicochemical parameters vectors had a significant correlation with the nMDS ordination? Is the origin of all plotted vectors in the 0,0 centered axes of the ordination of the samples? If so, in 4B it would appear to be a single group and not two. The view of both plots should show the 0,0 intersection of the NMDS1 and NMDS2 axes, including the scaling values and the labels.

The vectors are not necessarily significantly correlated with the nMDS ordination nor 0,0 centered. This is the default graphic representation of the software PRIMER we used. To the best of our knowledge, axes are usually not graphically represented in the nMDS plot. Moreover, we had already overlaid many physical-chemical and biological variables in these plots and adding also the axes would mean filling those plots with too many elements, rendering them too complicated to understand.

Table S2: The taxonomic identification of the taxa included in this table was done by classical taxonomy or by sequencing? the approach used should be mentioned in the legend of the table.

We integrated accordingly.

 Table S3: The legend starts "Diversity indices applied to MPB abundances in the two experiments" but I consider it'd be more appropriate "Diversity indices of MPB communities sampled in the two experiments"

We now changed and conformed both the captions of Table S2 and S3.

Tables S3 and S4: Were the indices calculated with about the same number of reads per sample? If the number of reads/individuals per sample varies by more than about 10%, the calculated parameters could not be comparable to draw conclusions.

Data obtained through classical taxonomy report all the cells that were actually present in the samples, the diversity indices rely on this kind of data to draw conclusions, therefore, we think that a rarefaction of the matrix wouldn’t be appropriate.

For the data obtained through metabarcoding, the different sequencing depth among samples might bias a little bit the indices results, therefore we now normalized the ASV table and re-run the analyses. As expected, the new results were consistent and comparable with the previous ones. We now additionally removed other 22 ASVs not univocally attributable to unicellular taxa/life-stages.

 Discussion

You refer to recent studies that claimed that the closure of MOSEs would only be effective for moderate sea level rises and that repeated and prolonged closures will be necessary and rapidly deplete oxygen level in the lagoon. Was that reducing in oxygen level observed/confirmed with these experiments? Or there are evidence published by other authors of the reduction in O2 concentrations since the implementation of MOSEs?

Yes, studies already exist about these predictions, we already mentioned them in the introduction and now also in the discussion. Our data confirmed O2 concentration reduction due to confinement, now we have added the results of the statistical test in the results section.

To “…limit oxygen and nutrients exchanges and favor the gradual sinking of the particulate matter from the water column toward the sediment (personal communication)”, there are some general results now to include in this manuscript or any personal communication to cite?

We corrected it in the text.

 You mention that “This rapid increase was likely ascribable to TOC and TN enrichment in the surface sediments (Fig. 4)…” but TN is not included in the Fig.4 and there is no explanation about that. In the October experiment, in that figure the increase seems to be also related to BPC but in July experiment it seems to be the contrary.

The reviewer is right, we removed TN and silt because highly correlated and therefore redundant to TOC and sand respectively; we now corrected the relative part in the methods section.

To “In contrast, metabarcoding can also quantify DNA residues from dead organisms…”, it can be avoided by working with cDNA obtained from extracted total RNA, instead of extracting total DNA. That way, only the active organisms synthesizing RNA will be detected, because RNA is less stable than DNA and no residues from dead cells will be recovered.

We thank the reviewer for the suggestion, the target of this study was to compare the classical taxonomy with the most-commonly used 18S rRNA gene sequencing. We now specified in the text that we refer only to 18S rRNA gene sequencing.

Please, reformulate the last part of the 4.2 metabarcoding section taking into account all the above commented. The last sentence should be modified to convey the idea that molecular tools based on the extraction of total DNA provide a more accurate picture of what is present in the community…

We are aware that many molecular tools are available nowadays and that, by employing several of them it is possible to obtain a comprehensive picture of the whole community; nevertheless, the classical taxonomy may be still very informative. We agree that in some cases molecular tools may provide a more accurate picture of what is present in the community, but sometimes, and especially with larger cells such as diatoms, the classical taxonomy is more informative particularly from a quantitative point of view. In presence of larger cells, the classical taxonomy is much more reliable in terms of absolute abundances, which are of paramount importance in ecological studies. Moreover, we observed that databases on microalgae from surface sediments obtained from metabarcoding are still poorly implemented and often give an assignment only at a higher hierarchical level (e.g., family or order). Therefore, we do not agree with the statement that molecular tools based on the extraction of total DNA provide a more accurate picture of what is present in the community, because we believe this is case-specific, and we do not wish to change this sentence.

 Instead of “Ecological valence of altered…” in the 4.3 section title “Ecological value of altered…” would be better.

We replaced the term “Ecological valence” with “Ecological aspects”.

Reviewer 3 Report

This manuscript presents results of a study focusing on the impact of artificial human constructions, i.e.  MOSE flood barriers on microphytobenthic community of the Venice lagoon, which in my opinion is an interesting and timely subject.

The methods decribed in the manuscript seem to be sound. I found the manuscript properly written and interesting.

Despite the overall good impression, the authors should consider a few issues for the future. I understand that this is the first work in this area, so in the future, this research will provide some directions on what parameters enrich the work, because it is impossible to do everything at once. The first thing that draws attention is the choice of a very short V9 marker. I don't mind that it was proposed in 2009 and there are newer ones, if they work then it's the authors' choice which one to choose. However, in the latest papers on metabarcoding, it is recommended that the V9 region be associated with the V4 region, making them an almost perfect pair. The next issue is the next step in community research based on RNA. Then, such a comparison of classical and molecular taxonomy would make more sense. I think the authors are aware of this because they point to the weakness of DNA-based research that also includes dead or falling from the water column organisms. The last issue is the poor correlation of abiotic parameters. We seem to have a classic case of a missing variable here, and authors may need to expand the research range in this direction.

The only technical note is to enlarge point B of Figure 1 because it is poorly visible.

I wish the authors further development of the research because the subject seems on time.

Despite some flaws, I recommend accepting this manuscript in its present form.

Author Response

We thank the reviewer for his/her observations and wishes. We below answer to his/her points.

- Despite the overall good impression, the authors should consider a few issues for the future. I understand that this is the first work in this area, so in the future, this research will provide some directions on what parameters enrich the work, because it is impossible to do everything at once. The first thing that draws attention is the choice of a very short V9 marker. I don't mind that it was proposed in 2009 and there are newer ones, if they work then it's the authors' choice which one to choose. However, in the latest papers on metabarcoding, it is recommended that the V9 region be associated with the V4 region, making them an almost perfect pair.

  • We thank the reviewer for the suggestion, which may be useful for future research on the subject. To be honest, choosing the right primer pair to characterise the enormous diversity of microbial communities is always a complex task. Clearly, we also had to face this issue. The primer pair was chosen in accordance with what had already been done in the Tara project (Alberti et al., 2017). In addition, the sequencing of the samples collected in 2019 was carried out at the end of the same year, and, once done using these primers, it was a natural choice to sequence the 2020 samples in the same way. To the best of our knowledge, the first paper suggesting the simultaneous use of both primers was published in April 2020 (Choi et al., 2020), thus after our first MiSeq run.

- The next issue is the next step in community research based on RNA. Then, such a comparison of classical and molecular taxonomy would make more sense. I think the authors are aware of this because they point to the weakness of DNA-based research that also includes dead or falling from the water column organisms.

  • We are aware of other molecular approaches. Nevertheless, since 18S rRNA gene sequencing is still the most widespread method for ecological studies, we were interested in comparing this particular technique with the classical taxonomy. We now clarified it better in the text.

- The last issue is the poor correlation of abiotic parameters. We seem to have a classic case of a missing variable here, and authors may need to expand the research range in this direction.

  • Our original dataset includes data relative to several abiotic parameters. As mentioned in the methods section, we performed a preliminary Spearman's correlation on the whole dataset, including TN, TOC, BPC and grain size fractions (later presented in the statistical analyses), temperature, salinity plus O2, CO2, NH4, NO2, NO3, SiO2, chlorophyll-a and phaeopigments concentrations. Only the most correlated variables were kept for further analyses.

Round 2

Reviewer 1 Report

Authors have revised the manuscript according to the reviewer's comments, and basically answered the reviewer's concerns. Hence, now I can recommend to accept the manuscript to be published.

Author Response

We thank the reviewer for endorsing our manuscript.

Reviewer 2 Report

The authors have made a great effort, the manuscript has greatly improved and the responses to my comments are appropriate and justified. However, I do not agree with some aspects, specially what refers to Cyanobacteria, but may be due to differences in criteria because I work mainly with prokaryotes. To explain myself, for me and also for other microbiologists, the identification and taxonomic classification of prokaryotes is not considered feasible by simply observing their morphology, much less their quantification. It might have been considered as such long ago, when cyanobacteria were called blue-green algae, but the prokaryotic domains Bacteria and Archaea have long been accepted and, in addition to their morphological characteristics, their functional characters are evaluated and/or their molecular phylogeny is analyzed for identification. In part, it is addressed, i.e., in doi: 10.1007/s10750-014-1971-9.

Relating to the last revision, I have marked some minor corrections on the file microorganisms-2189427-resubmitted-revised, as highlights with notes seen at opening them (attached) and hereby make some comments with respect to the last corrected version and your replies to my comments.

1-         One of your answers was “the information at T4 was still reliable also with two replicates only. These replicates were taken from two mesocosms with a sediment area of 0.785 m each, quite a representative portion of the lagoon. From each of the sampled sediment replicates, three counts were performed. In autumn the environmental conditions differed greatly from the summer ones, and the two obtained datasets were not comparable, in fact they were treated separately. On 4-6 December 2020, the MOSE system was closed for 48 hours nonstop, and this was the longest closure considering the closures performed till then. Usually the closures are much shorter (from a few hours to 24 hours) and the MOSE is activated only in very high tide conditions (105 cm), this can occur up to 4-5 times in each autumn month”. This information should be mentioned in the text in some way, either in methodology, in results, or in the introduction.

2- Regarding the results obtained from the nMDS ordination and BIOENV analysis, I meant to make it clear how those results were interpreted to predict the influence of environmental parameters on the microalgal community composition.

 3- The potential explanation “The PR2 database was chosen because it focuses mainly on unicellular eukaryotes, and it may not be particularly curated for certain taxa of multicellular organisms. At the same time, it is possible that the organisms in the sediment are less characterized from a genomic point of view than the planktonic ones” should be mentioned in the discussion.

 4- Also, it would be nice to include the comment “We further studied this not classified ASVs, which account for 48.4% of the initial ASVs. On average, these ASVs accounted for 13 ± 5.87% of the total counts in the samples. 781 were actually assigned to the Supergroup level of the PR2 database, divided as follows: Alveolata (115), Amoebozoa (125), Archaeplastida (43), Excavata (5), Hacrobia (4), Opisthokonta (246), Rhizaria (17), Stramenopiles (226)” in the results section.

 5- The information that “For the metabarcoding data, the n. of sequences/sample varied between 13 and 16% in the two experimental times of the first experiment and between 16 and 26% in the two experimental times of the second experiment” should also be included in the results section.

 6-You added the text “Our guiding questions and hypotheses were:…”. And hypotheses should be removed, as what follows are only your guiding questions. If you prefer to mention your hypothesis, those would be affirmative sentences referring to facts, as if the answer to both questions were "yes". For example, the first question without "does" and without the question mark: "The particular matter settling from the water column towards the sediment have a stimulatory effect, leading to an increase in MPB absolute abundances". That hypothesis, after the experiments and data analysis, can be confirmed (what already happened) or refused (if you hadn´t found differences along sampling times).

 7- In Methodology, Statistical analyses, it is written “Diversity analysis was applied to MPB abundances (genus and species or ASV level)…”. I consider this is not properly expressed and I would clarify the text in parenthesis. As I know, microscopy counts of cells and number of ASVs are two approaches that can be used to estimate abundances (number of individuals per mass or volume unit of sample). Then, the taxonomic identification to any rank level (phylum, family, genus, species) of the cells seen under the microscope or the ASVs obtained by sequencing ribosomal DNA amplicons is different from the abundances of organisms.

Alpha diversity analysis (what you refer to here as univariate analysis), consider how many different taxa has one sample (i.e. richness) and the abundance each taxa has in that sample (i.e. number of individuals counted, ASVs sequenced, OTUs defined). Then, the taxonomic rank/identification considered to estimate the abundance of each taxa can be selected and could be to phylum level, class level, genus level, species level (or any rank) or ASV level or OTU level when the approach was sequencing rRNA gene amplicons and the OTUs/ASVs were not collapsed to a certain taxonomic rank. In Tables S1 and S2, the taxonomic identifications shown for cells observed and ASVs are to the species level. According to the legends of Tables S5 and S6, I understand that diversity indices were calculated using the MPB abundances at species level (cells cm-3) and ASV level (number of ASVs). If this is not right, please make it clearer in the legends of the corresponding supplementary tables.

 8- In addition, and as the secondary aim of this manuscript is to compare both taxonomic classification approaches, I consider it would be useful to include the concepts you provided in your answer to my question: “We agree that in some cases molecular tools may provide a more accurate picture of what is present in the community, but sometimes, and especially with larger cells such as diatoms, the classical taxonomy is more informative particularly from a quantitative point of view. In presence of larger cells, the classical taxonomy is much more reliable in terms of absolute abundances, which are of paramount importance in ecological studies. Moreover, we observed that databases on microalgae from surface sediments obtained from metabarcoding are still poorly implemented and often give an assignment only at a higher hierarchical level (e.g., family or order).

In the Discussion section, at the last part of the “Microphytobenthic community through metabarcoding” item, it would be useful for the readers to understand your criteria, to add a paragraph between “…biodiversity. These results…”, like “… biodiversity. Moreover, …. These results…” and after moreover include the concepts you referred to in your answer.

 9- About the comment I made regarding the Figure 4, the fitting of environmental variables to an ordination plot is somehow a regression analysis and has a significance or a coefficient. Some randomly selected plots can be found in Google images.

 10- Regarding Figure 6, my question was about the total number of sequences for each replicate. Your answer was “As mentioned above, these are biological replicates, and the variability is within the range common for this kind of studies. For the metabarcoding data, the n. of sequences/sample varied between 13 and 16% in the two experimental times of the first experiment and between 16 and 26% in the two experimental times of the second experiment.” The observed variability, over 10%, is not the common range for other metabarcoding studies. That these are biological replicates and the variation (% range) of the two experiments should be mentioned in the results section.

  11- The supplementary information as it is inside the word file is not properly formatted. I was sent an excel file, where the tables and figures look better, but the legends are not included. Even when the files of supplementary information are uploaded as the authors send them, they should be clear and understandable for the readers. I assume you will upload as supplementary information the excel file. If not, and the legends will be included after the link to download the files, make sure that all will be needed for readers to understand the tables/figures, is mentioned.

Author Response

Comments and Suggestions for Authors

The authors have made a great effort, the manuscript has greatly improved and the responses to my comments are appropriate and justified. However, I do not agree with some aspects, specially what refers to Cyanobacteria, but may be due to differences in criteria because I work mainly with prokaryotes. To explain myself, for me and also for other microbiologists, the identification and taxonomic classification of prokaryotes is not considered feasible by simply observing their morphology, much less their quantification. It might have been considered as such long ago, when cyanobacteria were called blue-green algae, but the prokaryotic domains Bacteria and Archaea have long been accepted and, in addition to their morphological characteristics, their functional characters are evaluated and/or their molecular phylogeny is analyzed for identification. In part, it is addressed, i.e., in doi: 10.1007/s10750-014-1971-9.

We thank the Reviewer for this comment, in our next paper we will avoid including Cyanobacteria and will focus on diatoms, only. As we already explained, in this paper we focused on the entire microphytobenthic community, therefore Cyanobacteria were also considered, though only a very few specimens were encountered and identified.

Relating to the last revision, I have marked some minor corrections on the file microorganisms-2189427-resubmitted-revised, as highlights with notes seen at opening them (attached) and hereby make some comments with respect to the last corrected version and your replies to my comments.

The editor didn’t provide us with this file, since direct comments on a PDF file are not set out for the revision process. We got it only at the end of the revision time and applied the minor corrections marked therein.

1- One of your answers was “the information at T4 was still reliable also with two replicates only. These replicates were taken from two mesocosms with a sediment area of 0.785 m2 each, quite a representative portion of the lagoon. From each of the sampled sediment replicates, three counts were performed. In autumn the environmental conditions differed greatly from the summer ones, and the two obtained datasets were not comparable, in fact they were treated separately. On 4-6 December 2020, the MOSE system was closed for 48 hours nonstop, and this was the longest closure considering the closures performed till then. Usually the closures are much shorter (from a few hours to 24 hours) and the MOSE is activated only in very high tide conditions (105 cm), this can occur up to 4-5 times in each autumn month”. This information should be mentioned in the text in some way, either in methodology, in results, or in the introduction.

We have added a few sentences in the text giving this information.

2- Regarding the results obtained from the nMDS ordination and BIOENV analysis, I meant to make it clear how those results were interpreted to predict the influence of environmental parameters on the microalgal community composition.

It is very difficult to “predict” the influence of environmental parameters on the microalgal community composition. There are too many physical-chemical variables to consider. To do this, we should build a mathematical model to obtain a reliable outcome. Otherwise, these would be only inferences, therefore we prefer interpreting our findings without any kind of predictions.

 3- The potential explanation “The PR2 database was chosen because it focuses mainly on unicellular eukaryotes, and it may not be particularly curated for certain taxa of multicellular organisms. At the same time, it is possible that the organisms in the sediment are less characterized from a genomic point of view than the planktonic ones” should be mentioned in the discussion.

The first sentence was just the explanation to the Reviewer’s previous comment.

Since the focus of this study is about unicellular organisms and multicellular ones are not considered neither mentioned, we believe including the first sentence would be confusing and misleading. We have, however, included the second sentence in the text.

 4- Also, it would be nice to include the comment “We further studied this not classified ASVs, which account for 48.4% of the initial ASVs. On average, these ASVs accounted for 13 ± 5.87% of the total counts in the samples. 781 were actually assigned to the Supergroup level of the PR2 database, divided as follows: Alveolata (115), Amoebozoa (125), Archaeplastida (43), Excavata (5), Hacrobia (4), Opisthokonta (246), Rhizaria (17), Stramenopiles (226)” in the results section.

We have added few details in the relative results section.

 5- The information that “For the metabarcoding data, the n. of sequences/sample varied between 13 and 16% in the two experimental times of the first experiment and between 16 and 26% in the two experimental times of the second experiment” should also be included in the results section.

We have added it accordingly.

 6-You added the text “Our guiding questions and hypotheses were:…”. And hypotheses should be removed, as what follows are only your guiding questions. If you prefer to mention your hypothesis, those would be affirmative sentences referring to facts, as if the answer to both questions were "yes". For example, the first question without "does" and without the question mark: "The particular matter settling from the water column towards the sediment have a stimulatory effect, leading to an increase in MPB absolute abundances". That hypothesis, after the experiments and data analysis, can be confirmed (what already happened) or refused (if you hadn´t found differences along sampling times).

We have corrected accordingly.

 7- In Methodology, Statistical analyses, it is written “Diversity analysis was applied to MPB abundances (genus and species or ASV level)…”. I consider this is not properly expressed and I would clarify the text in parenthesis. As I know, microscopy counts of cells and number of ASVs are two approaches that can be used to estimate abundances (number of individuals per mass or volume unit of sample). Then, the taxonomic identification to any rank level (phylum, family, genus, species) of the cells seen under the microscope or the ASVs obtained by sequencing ribosomal DNA amplicons is different from the abundances of organisms.

We have better clarified the text in parenthesis. “Diversity analysis was applied to MPB data i.e. abundances at genus and species level obtained by classical microscopy or ASV level obtained through molecular tools…

Alpha diversity analysis (what you refer to here as univariate analysis), consider how many different taxa has one sample (i.e. richness) and the abundance each taxa has in that sample (i.e. number of individuals counted, ASVs sequenced, OTUs defined). Then, the taxonomic rank/identification considered to estimate the abundance of each taxa can be selected and could be to phylum level, class level, genus level, species level (or any rank) or ASV level or OTU level when the approach was sequencing rRNA gene amplicons and the OTUs/ASVs were not collapsed to a certain taxonomic rank. In Tables S1 and S2, the taxonomic identifications shown for cells observed and ASVs are to the species level. According to the legends of Tables S5 and S6, I understand that diversity indices were calculated using the MPB abundances at species level (cells cm-3) and ASV level (number of ASVs). If this is not right, please make it clearer in the legends of the corresponding supplementary tables.

As it can be appreciated from Table S1, not all the organisms could be identified at species level, therefore, as we specified in the methods, for classical taxonomy data, also the genus level was included.

For the metabarcoding data, as reported in the methods, the level is always the ASV even when unclassified to the species level.

We have now added the detail in the legend of Table S5, in the legend of Table S6 the ASV level was already specified.

 8- In addition, and as the secondary aim of this manuscript is to compare both taxonomic classification approaches, I consider it would be useful to include the concepts you provided in your answer to my question: “We agree that in some cases molecular tools may provide a more accurate picture of what is present in the community, but sometimes, and especially with larger cells such as diatoms, the classical taxonomy is more informative particularly from a quantitative point of view. In presence of larger cells, the classical taxonomy is much more reliable in terms of absolute abundances, which are of paramount importance in ecological studies. Moreover, we observed that databases on microalgae from surface sediments obtained from metabarcoding are still poorly implemented and often give an assignment only at a higher hierarchical level (e.g., family or order).”

We have included these concepts in the discussion.

In the Discussion section, at the last part of the “Microphytobenthic community through metabarcoding” item, it would be useful for the readers to understand your criteria, to add a paragraph between “…biodiversity. These results…”, like “… biodiversity. Moreover, …. These results…” and after moreover include the concepts you referred to in your answer.

We now edited a bit this part.

 9- About the comment I made regarding the Figure 4, the fitting of environmental variables to an ordination plot is somehow a regression analysis and has a significance or a coefficient. Some randomly selected plots can be found in Google images.

Below we report the correlation coefficients of the environmental variables overlaid on the nMDS plots.

sand

clay

TOC

C-H2O

C-EDTA

C-PRT

C-LIP

SUMMER 2019

MDS1

0.181

-0.608

-0.206

-0.290

0.518

0.038

0.100

MDS2

0.205

-0.001

0.158

-0.314

0.143

-0.203

-0.081

AUTUMN 2020

MDS1

0.018

-0.085

-0.528

-0.616

-0.496

-0.235

-0.379

MDS2

-0.430

0.190

-0.088

-0.131

0.329

0.176

0.105

 10- Regarding Figure 6, my question was about the total number of sequences for each replicate. Your answer was “As mentioned above, these are biological replicates, and the variability is within the range common for this kind of studies. For the metabarcoding data, the n. of sequences/sample varied between 13 and 16% in the two experimental times of the first experiment and between 16 and 26% in the two experimental times of the second experiment.” The observed variability, over 10%, is not the common range for other metabarcoding studies. That these are biological replicates and the variation (% range) of the two experiments should be mentioned in the results section.

In our knowledge, our range of variability falls within what is commonly observed in similar kind of studies (https://doi.org/10.3897/mbmg.5.71107). Our methods are consistent for all the replicates and we applied both quality check and sequences filtering, therefore, the variability we observed between the replicates is only ascribable to the biological diversity.

We have added this information in the results section.

  11- The supplementary information as it is inside the word file is not properly formatted. I was sent an excel file, where the tables and figures look better, but the legends are not included. Even when the files of supplementary information are uploaded as the authors send them, they should be clear and understandable for the readers. I assume you will upload as supplementary information the excel file. If not, and the legends will be included after the link to download the files, make sure that all will be needed for readers to understand the tables/figures, is mentioned.

The Excel file was included only because of the visualization issues with the Word file between different OSs. The legends are included in the Word file as required by the journal. We’ll take care that everything will be uploaded and published in a clear and available format. We thank the Reviewer for their concern.
